# Norepinephrine transporter defects lead to sympathetic hyperactivity in Familial Dysautonomia models

Hsueh-Fu Wu[1,2], Wenxin Yu [3], Kenyi Saito-Diaz [1], Chia-Wei Huang[2,4], Joseph Carey[5], Frances Lefcort[5], Gerald W. Hart[2,4], Hong-Xiang Liu[3] & Nadja Zeltner [1,2,6] ✉

Familial dysautonomia (FD), a rare neurodevelopmental and neurodegenerative disorder affects the sympathetic and sensory nervous system. Although almost all patients harbor a mutation in ELP1, it remains unresolved exactly how function of sympathetic neurons (symNs) is affected; knowledge critical for understanding debilitating disease hallmarks, including cardiovascular instability or dysautonomic crises, that result from dysregulated sympathetic activity. Here, we employ the human pluripotent stem cell (hPSC) system to understand symN disease mechanisms and test candidate drugs. FD symNs are intrinsically hyperactive in vitro, in cardiomyocyte co-cultures, and in animal models. We report reduced norepinephrine transporter expression, decreased intracellular norepinephrine (NE), decreased NE re-uptake, and excessive extracellular NE in FD symNs. SymN hyperactivity is not a direct ELP1 mutation result, but may connect to NET via RAB proteins. We found that candidate drugs lowered hyperactivity independent of ELP1 modulation. Our findings may have implications for other symN disorders and may allow future drug testing and discovery.

Familial Dysautonomia (FD) is a complex, early-onset, genetic disorder that mainly affects the development of the sensory and sympathetic nervous system (SNS) within the peripheral nervous system (PNS), leading to reduced numbers of these neurons. It also leads to neuro-degeneration of these cells. 99.5% of all FD patients harbor a nonsense RNA splicing mutation in the gene *ELP1* (formerly called *IKBKAP*[1], which encodes the protein Elongator complex protein 1 (ELP1). ELP1 is the structural component of the Elongator complex, which is directly involved in tRNA modification[2] and is essential for normal translation, elongation and other cellular functions[3], and thus can be found in most cells of the body. While FD patients also suffer deficits in the central nervous system (CNS, particularly in the eye), due to the critical role of the autonomic nervous system (ANS) in maintaining homeostasis, the

altered function of the ANS is the most life-threatening defect in FD. It was proposed that *ELP1* defects mainly affect particularly long transcripts with an AA nucleotide bias and that many PNS critical genes fall into this category[4]. FD patients suffer from a variety of symptoms, including difficulties swallowing and regulating heart rate and blood pressure[5]. Dysautonomic/adrenergic crises are particularly disabling episodes of tachycardia, arterial hypertension, nausea/vomiting/retching, and behavioral changes reminiscent of anxiety attacks[6]. Crisis can be frequent (daily), increase with age[7], lead to hospitalization and contribute significantly to mortality[8]. Interestingly, crises are triggered by emotional or physical stress and associated with anxiety that can escalate into phobias[9]. Uncontrollable sympathetic neuron (symN) over-stimulation and associated prolonged catecholamine secretion

[1]Center for Molecular Medicine, University of Georgia, Athens, GA, USA. [2]Department of Biochemistry and Molecular Biology, University of Georgia, Athens, GA, USA. [3]Regenerative Bioscience Center, Department of Animal and Dairy Science, College of Agricultural and Environmental Sciences, University of Georgia, Athens, GA, USA. [4]Complex Carbohydrate Research Center, University of Georgia, Athens, GA, USA. [5]Department of Microbiology and Cell Biology, Montana State University, Bozeman, MT, USA. [6]Department of Cellular Biology, University of Georgia, Athens, GA, USA. ✉e-mail: nadja.zeltner@uga.edu

into the blood stream caused by afferent baroreflex failure has been suggested as the mechanisms mediating FD crisis[10]. This indeed explains the increase in blood pressure during crisis. Nevertheless, the fact that non-FD patients with baroreflex failure mainly suffer from unstable blood pressure, but do not experience many of the other aspects of the FD dysautonomic crisis, suggests that the loss of central autonomic control and baroreflex defects may not be the only factor that leads to the crisis. For example, vomiting/retching episodes, characteristic for FD dysautonomic crisis, do not occur in baroreflex failure patients. In FD, it was suggested that vomiting/retching is a result of excessive dopamine (DA) in the blood stream[11]. Using microneurography, it was also shown in FD that sympathetic nerve activity is spontaneously desynchronized, where the neurons do not fire with the characteristic bursting pattern in between heart beats, rather tending to fire continuously during periods of emotional arousal[12]. Therefore, we hypothesize that intrinsic factors within symNs in FD contribute to triggering dysautonomic crisis.

Reduced symN numbers and overall sympathetic ganglia size reduction has been reported in FD patients[13] and in FD mouse models[14-17]. Such observations are based on both developmental defects[14] and degeneration[13,14]. While several studies in mouse neurons have revealed that neurons in FD have impairments in mitochondrial function[18], elevated reactive oxygen species, and cytoskeleton issues[19]; that their proteome and transcriptome is affected[4], and they have faulty nerve growth factor (NGF) retrograde transport and signaling[20], no study on neuronal activity has been reported, particularly not in human cells. Loss, hypo- or hyperactivity of symNs leads to a variety of human disorders and intrinsic neural hyperactivity has been proposed as the cause of dysfunction and ensuing degeneration in several disorders, including amyotrophic lateral sclerosis (ALS)[21] and Alzheimer's disease[22]. Hyperactivity is further linked with aberrant calcium homeostasis[23] and norepinephrine (NE) levels[24]. Therefore, we aim to investigate symN-specific phenotypes in FD, with the long-term goal to provide knowledge for drug development for FD as well as other symN disorders.

Assessments of symN function in the clinic are mostly based on indirect measures, such as blood pressure, heart rate and blood glucose levels[25], and the lack of availability of primary sympathetic tissue for research makes it challenging to address neural intrinsic dysfunctions. Thus, it remains to be shown that mechanistic insights into FD gained from mouse studies are present in human cells as well. The human pluripotent stem cell (hPSC) technology is ideal to address such shortcomings, as it allows the generation of near unlimited numbers of human, patient-derived symNs and their observation and manipulation in the dish[26]. In fact, FD was one of the first diseases modeled with the hPSC technology[27,28]. Furthermore, the technology was used to confirm and recapitulate differing phenotypes between FD patients presenting with mild and more severe symptoms and to correlate this variance to potential modifier mutations in three patients[29], though these findings have to be further confirmed in the larger FD population.

Here, we employ our well-established symN differentiation protocol[30] to study functional phenotypes in FD symNs, using FD patient- and healthy control-derived hPSCs. We recapitulate developmental defects in the sympathetic lineage and reveal spontaneous, intrinsic hyperactivity of FD symNs. We identify a defective norepinephrine autoregulatory pathway that underlies the hyperactivity phenotype. We show that ELP1 may connect to NET defects and hyperactivity through RAB proteins involved in cellular trafficking. Lastly, we show that this model system is sensitive in confirming clinical drugs and potential drug compounds that can relieve symN hyperactivity. Together, our results reveal symN hyperactivity as a novel pathology in FD, and we provide a novel, human drug testing and screening platform for symN-modulating compounds.

## Results

### Developmental phenotypes in the FD sympathetic lineage

To model symN defects in FD, we needed an efficient and reproducible differentiation protocol to derive pure symNs from hPSCs. Although there are several published symN differentiation strategies[30-34], we first further optimized our previous feeder-free, chemically defined protocol (Supplementary Fig. 1a). Neural crest cells (NCCs) at day 10 are derived at an efficiency of >90%[30], thus a time-consuming flow cytometry (FACS) purification step can be omitted. SymNs from day 14 on express sympathoblast genes including PHOX2B, ASCL1, and HAND2 (Supplementary Fig. 1b) and express HOX genes (HOX 5–9) indicating their trunk-like identity (Supplementary Fig. 1c). They display typical PNS ganglia-like morphology (Supplementary Fig. 1d). Mature marker genes are expressed from day 20 on, on the mRNA and the protein level (Supplementary Fig. 1e–h), including genes important in symN signal transduction CHRNA3, CHRNB4 and VMAT1, autoregulation $\alpha_2/\beta_2$ARs and NET and NE synthesis/metabolism TH, DBH, AAAD and MOA-A. We also confirmed NE production in symNs (Supplementary Fig. 1h). To improve technical ease of the protocol, we show that both human embryonic stem cell (hESC)- and induced pluripotent stem cell (iPSC)-derived NCCs can be cryopreserved at day 10, followed by differentiation into symNs, without compromising the neuron quality (Supplementary Fig. 1i, j).

Next, we sought to improve the purity of differentiated symNs to prevent possible variations due to contaminating non-neuronal cells during phenotype identification. Our original protocol yielded about 45% neurons, of which over 90% were TH+ symNs (Supplementary Fig. 2a, gray bars). To eliminate non-neuronal lineages, we treated day 20 symNs with aphidicolin (Aphi), a cell cycle inhibitor commonly used in primary peripheral neuron cultures[35,36], for 10 days. After day 30, Aphi-treated symNs show significantly improved purity (~75% of all cells are TH+ symNs), while neural specificity is not affected (Supplementary Fig. 2a, b, red bars). Proliferating Ki67+ cells and SOX10+ NCCs are dramatically diminished after Aphi treatment (Supplementary Fig. 2c). Furthermore, we analyzed electrical activity of symNs after day 20 using multi-electrode array (MEA) (Supplementary Fig. 2d). Neural spikes represent functionality and neural bursts represent neuronal maturity, which increases over time (Supplementary Fig. 2e, f). In FD, the potentially defective interplay between the sympathetic and parasympathetic division remains unclear and somewhat controversial[37-41], thus it is important to verify that our differentiation protocol is free of parasympathetic neurons (parasymN). We did not detect any typical parasymN markers, i.e., 90% of all neurons are TH+ (which is not expressed in parasymNs, Supplementary Fig. 2a), the parasymN marker choline acetyltransferase (ChAT) was absent (Supplementary Fig. 2f), and acetylcholine (ACh) release was ~7 times reduced compared to NE (Supplementary Fig. 2g). ACh is the primary parasymN neurotransmitter, however few symNs release ACh as well. Thus, this protocol yields high numbers of pure and functional symNs within a shortened time-period.

With this differentiation protocol in hand, we next aimed to assess phenotypes in the symN lineage in FD. We employed the following hPSC lines: iPSCs from FD patients (iPSC-FD-S2 and iPSC-FD-S3) and healthy control subjects (iPSC-ctrl-C1), and healthy hESCs (hESC-ctrl-H9 (WA09), Supplementary Fig. 3a). These lines were previously established, well characterized and employed in disease modeling[29]. We first assessed disease phenotypes throughout the developmental stages of NCCs, sympathoblasts and symNs in FD (Fig. 1a). NCCs can be identified in condensed dark "ridge" structures (Fig. 1b, arrows), which are SOX10+ and correlate with CD49D[30,42] staining on day 10 (Fig. 1b–e). In accordance with previous findings[29], control NCCs (iPSC-ctrl-C1 and hESC-ctrl-H9) showed higher differentiation efficiency (~90%) compared to FD NCCs (iPSC-FD-S2 and iPSC-FD-S3, ~30%) evident by higher SOX10+ NCC ridge coverage (Fig. 1b, c) and higher CD49D expression (by

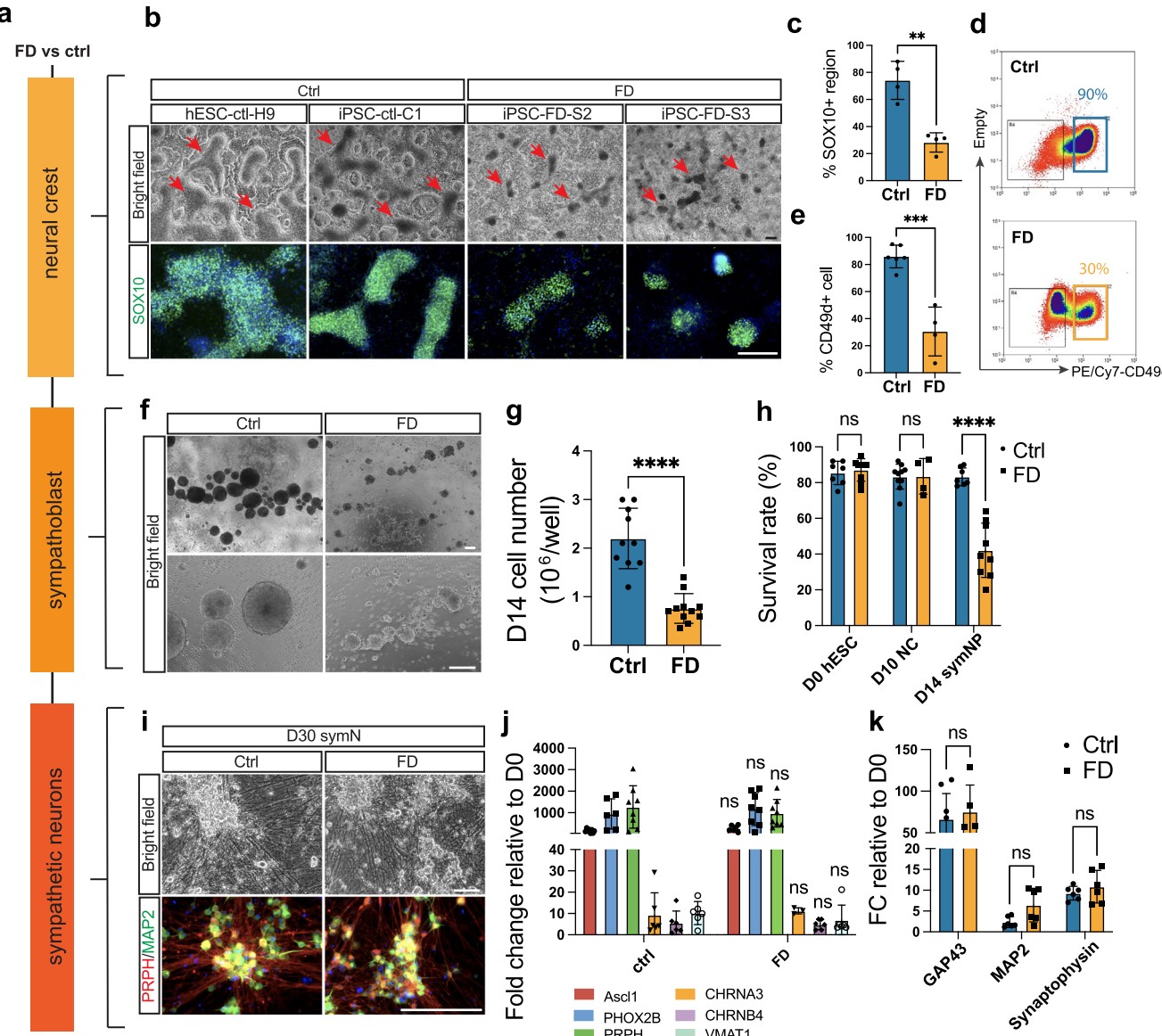

**Fig. 1 | Developmental phenotypes in Familial Dysautonomia (FD) sympathetic neurons (symNs). a** Schematic workflow of developmental comparisons during symN differentiation in control (ctrl) and FD groups. **b** Comparison of neural crest cell (NCC) differentiation on day 10. Red arrows indicate the NCC "ridge" structures. Immunostaining in bottom row shows SOX10 (green) positive NCC ridges at high scale. hESC-ctrl-H9 = human embryonic stem cells, WA09, hiPSCs = human induced pluripotent stem cells, C1 = healthy ctrl, S2, S3 = FD patients. **c** Quantification of SOX10⁺ cell area in (**b**). $n = 4$ biological replicates. Two-tailed Student's t-test. Error bar represents the SEM. **$p < 0.01$. **d** Day 10 NCC number is analyzed by FACS using PE/Cy7-CD49d. Blue rectangle in ctrl and yellow in FD gate CD49d⁺ fractions. **e** Quantification of CD49d⁺ cell number by FACS. $N = $ (ctrl: 6, FD: 4) biological replicates. Two-tailed Student's t-test. Error bar represents the SEM. ***$p < 0.001$. **f** Representative sympathoblast spheroids are compared on day14. **g** Day 14 total cell numbers after dissociation is compared. $N = $ (ctrl: 10, FD: 11) biological replicates. Two-tailed Student's t-test. Error bar represents the SEM. ****$p < 0.0001$. **h** Survival rates after dissociation are quantified and compared at

each stage. SymNP = sympathoblast. $N = $ (D0-ctrl: 7, FD: 9; D10-ctrl: 10, FD: 4; D14-ctrl: 7, FD:9) biological replicates. Two-tailed multiple unpaired Student's t-test. Error bar represents the SEM. ****$p < 0.0001$. **i** Representative differentiated symNs are compared after day 30 in bright field (top row) and PRPH (red)/MAP2 (green) staining (bottom row). **j** RT-qPCR analysis shows similar expressions of symN markers between control and FD. $N = $ (ASCL1-ctrl: 8, FD: 6; PHOX2B-ctrl: 6, FD: 8; PRPH: 8; CHRNA3-ctrl: 6, FD: 3; CHRNB4: 6; VMAT1: 6) biological replicates. Two-tailed multiple unpaired Student's t-test. Error bar represents the SEM. **k** Gene expression of neurite growth markers are further examined using RT-qPCR and no significant difference is found. $N = $ (GAP43-ctrl: 8, FD: 5; MAP2 & SYNAPTOPHYSIN: 6) biological replicates. Two-tailed multiple unpaired Student's t-test. Error bar represents the SEM. Scale bars represent 200 µm. In **c**, **g**, and **h**, data from hESC-ctrl-H9 and iPSC-ctrl-C1 are pooled as control; data from iPSC-FD-S2 and iPSC-FD-S3 are pooled as FD. See also Supplementary Figs. 1–4. Source data are provided as a Source Data file.

FACS, Fig. 1d, e). Such a ratio of control to FD NCC reduction (90–30%) correlates with the reduction of sympathetic ganglia volume described in FD patients[13] and mice[14,15]. From here on, we combined data for iPSC-ctrl-C1 and hESC-ctrl-H9 and called it ctrl, as well as iPSC-FD-S2 and iPSC-FD-S3 and called it FD as indicated in the figure legends. To answer the question, whether in FD the remaining 60% of cells die or differentiate into another cell type, we assessed

overall cell numbers. We found that at day 10 there is no difference in cell numbers (Supplementary Fig. 4a), which is corroborated by similar staining for AP2a (Supplementary Fig. 4b), suggesting that FD iPSCs may differentiate into other cell types, likely non-neural ectoderm, rather than die. In addition, using RT-qPCR analysis, we found significant expression of SIX1⁺/EYA1⁺ placode contaminants (Supplementary Fig. 4c). However, we previously showed that non-

NCCs do not aggregate into neural spheroids or survive[30], and thus are lost upon further differentiation.

Next, we compared control and FD NCCs during sympathoblast specification in spheroids from day 10–14 (Fig. 1f). Despite replating equal NCC numbers at day 10 in control and FD iPSCs, we observed smaller size as well as lower total cell numbers in FD compared to control spheroids by day 14 (Fig. 1f, g). We also noticed compromised integrity in FD spheroids, forming less compact, smooth, and more irregular-shaped aggregates (Fig. 1f). Additionally, we compared the survival rate between control and FD cells after dissociation at day 0, 10 and 14 and show that survival in FD drops at the sympathoblast stage, where symNs are specified (Fig. 1h). Previous studies using an FD mouse model identified decreased neural survival in E12.5 sympathetic ganglia on top of reduced ganglion size, while NCC survival was intact[15]. Our observations, therefore, capture the sympathetic defects at early developmental stages. Finally, we assessed symN differentiation upon dissociation and equal cell number replating of day 14 sympathoblasts. We found that, at day 30 both control and FD generate normal looking symNs with similar morphology and distribution (Fig. 1i). RT-qPCR analysis showed no significant differences of symN markers (Fig. 1j). SymN morphology was further examined using axonal marker GAP43, dendritic marker MAP2 and synaptic marker synaptophysin, and no difference was found (Fig. 1i, k). Together, our results suggest that in FD both the NCC and sympathoblast numbers are reduced, but the remaining progenitors are still capable of generating symNs. These findings are supported by our previous work[29] and other's works[13,43,44].

## FD sympathetic neurons are spontaneously hyperactive

After differentiation, FD symNs showed no appreciable morphological difference, thus, we next asked if the symNs that do develop in FD are functional. Utilizing MEA, we found that FD symNs are spontaneously hyperactive starting at day 25 (Fig. 2a). To investigate the possibility that the difference found at a single timepoint is because of varying differentiation progression/maturation of each iPSC line, MEA recording was performed in a time course from day 20 to 60. We found that compared to control, FD symNs are spontaneously hyperactive throughout their maturation and that up to day 60, control neurons never reach the spike frequency of FD (Fig. 2b), excluding the concern of differential maturation speed. Our finding of hyperactivity in FD symNs was further confirmed by increased expression of c-Fos, a gene that represents symN activity[45] (Fig. 2c), neuropeptide Y, a factor that will be upregulated following symN activation[46] (Fig. 2d) and corticotropin releasing hormone receptor 1 and 2 (CRHR1/2), the receptors for stress-induced CRH[47] (Fig. 2e), all highlighting increased neuronal activity and heightened stress responsiveness. Additionally, we confirmed FD symN hyperactivity by monitoring spontaneous calcium ($Ca^{2+}$) dynamics. Using the $Ca^{2+}$ probe Fluo-4, we observed that FD symNs have higher $Ca^{2+}$ influx activity than healthy control symNs during recording[48] (Fig. 2f), reflecting the hyperactive state in FD symNs. In order to understand whether the hyperactivity phenotype is a PNS-specific feature, we differentiated CNS cortical neurons from control and FD iPSCs using previously described protocols[49,50] and assessed their electric activity in parallel. We did not detect significant differences either in morphology (Fig. 2g) or mean firing rate (Fig. 2h) at or before day 45. These observations support that symN hyperactivity is a critical PNS phenotype in FD.

We next assessed the effect of hyperactive symNs on regulating their target tissue in FD. We differentiated cardiomyocytes[51,52] (CMs) from healthy hESC-ctrl-H9 and/or iPSC-ctrl-C1 lines, that started beating at day 7 (Supplementary Fig. 5a and Supplementary Video 1), fired cardiac action potentials from day 10 on (via MEA, Supplementary Fig. 5b) and expressed specific markers cTnT and NKX2.5 at day 15 (Supplementary Fig. 5c). To mimic the SNS-cardiac axis, we created a co-culture consisting of day 15 CM precursors and dissociated day

14 symNs in MEA plates (Fig. 2i). Innervation of cardiomyocytes by symNs can be observed one week later (Supplementary Fig. 5d), where typical neuro-cardiac muscle junction nodes can be observed (Supplementary Fig. 5d′–d′′′′) in stainings with both symN (PRPH, synaptophysin (SYP)) and CM (a-ACTININ) markers. CM beating rate increased when CMs were cultured with conditional medium from symNs, as well as in the co-culture with symNs. It could further be augmented when symNs were stimulated with nicotine (Supplementary Fig. 5e), indicating the functionality of the co-culture system. With this powerful tool, we assessed FD symN regulation of its cardiomyocyte target tissue. As shown previously[33], the co-culture matured and thus increased the beating rate even in control; however, it significantly increased CM beating rate in the FD symN co-culture (Fig. 2j). Together, we show that symNs in FD are spontaneously hyperactive which leads to increased target tissue activation, potentially paralleling FD patients heart rate instability[53].

## FD sympathetic neuron hyperactivity is conserved in mouse models

Next, we sought to corroborate our findings of symN hyperactivity in FD in vivo. Previous reports in various FD mouse models have shown symN loss at the embryonic stage, smaller sympathetic ganglion size and defective target innervation[14,15,19]. SymN activity however, has not been investigated previously to our knowledge. Here, we used two FD mouse models, the wnt1-cre; elp1[LoxP/LoxP] conditional knock-out (CKO) model[14] and the sox10-cre; elp1[LoxP/LoxP] CKO model[54], both of which delete elp1 expression in the neural crest cell lineage (Supplementary Fig. 6a). Most FD CKO mice, including ours, are embryonic lethal[15,19], and the few surviving pups are of smaller stature compared to control (Fig. 3a). We aimed at focusing on symN activity and thus chose E14.5, the embryonic stage where the superior cervical ganglia (SCG) formed recently[55–59] and their size is not different between control and FD yet. Previous reports have shown that the gross loss of progenitor cell mass in sympathetic ganglia begins before E17.5[14]. Accordingly, we did not observe an embryo body size difference between FD and control (Fig. 3b, c). We dissected the SCG (Supplementary Fig. 6b), dissociated the symNs and cultured them for 7 days, while measuring neural activity via MEA (Fig. 3d). The SCG were similar in size (Fig. 3e) and the symNs expressed the appropriate markers (Fig. 3f). SCGs from control and FD mice were plated evenly in MEA dishes (Fig. 3h, top) and mouse Elp1 was indeed knocked-out in FD, as no Elp1 protein was detectable in the FD cultures (Fig. 3h, bottom). FD and control symNs showed similar neurite development (Fig. 3h). We then compared symN activity from control and CKO mice for one week, showing that FD symNs fire spontaneously at a higher rate compared to WT neurons (Fig. 3i, j), consistent with our findings in the iPSC-based symN model. Finally, we found that SCG neurons start to degenerate from day 7 in vitro, supporting the notion that hyperactivity is detrimental, as seen in other systems[21] (Supplementary Fig. 6c). This data supports the finding that sympathetic neurons are spontaneously hyperactive in FD, both in hPSC-derived symNs and as well as in symNs derived from two FD mouse models.

## Defects in norepinephrine transporter underlie hyperactivity in FD

We next aimed at a better understanding of the molecular mechanism underlying symN hyperactivity in FD. 99.5% of all FD patients, including the patient's cells analyzed here[29], harbor a homozygous mutation in ELP1. To examine whether hyperactivity is the direct consequence of the ELP1 mutation, we first performed symN differentiations using an ELP1 rescued line, iPSC-EPL1rescued-T6, in which the ELP1 mutation is heterozygously corrected by CRISPR-cas9 from the iPSC-FD-S2 line[29]. Surprisingly, MEA recording over time revealed that iPSC-EPL1rescued-T6 symNs remain mostly hyperactive compared to control (Supplementary Fig. 7a). Since this was a somewhat puzzling result, we

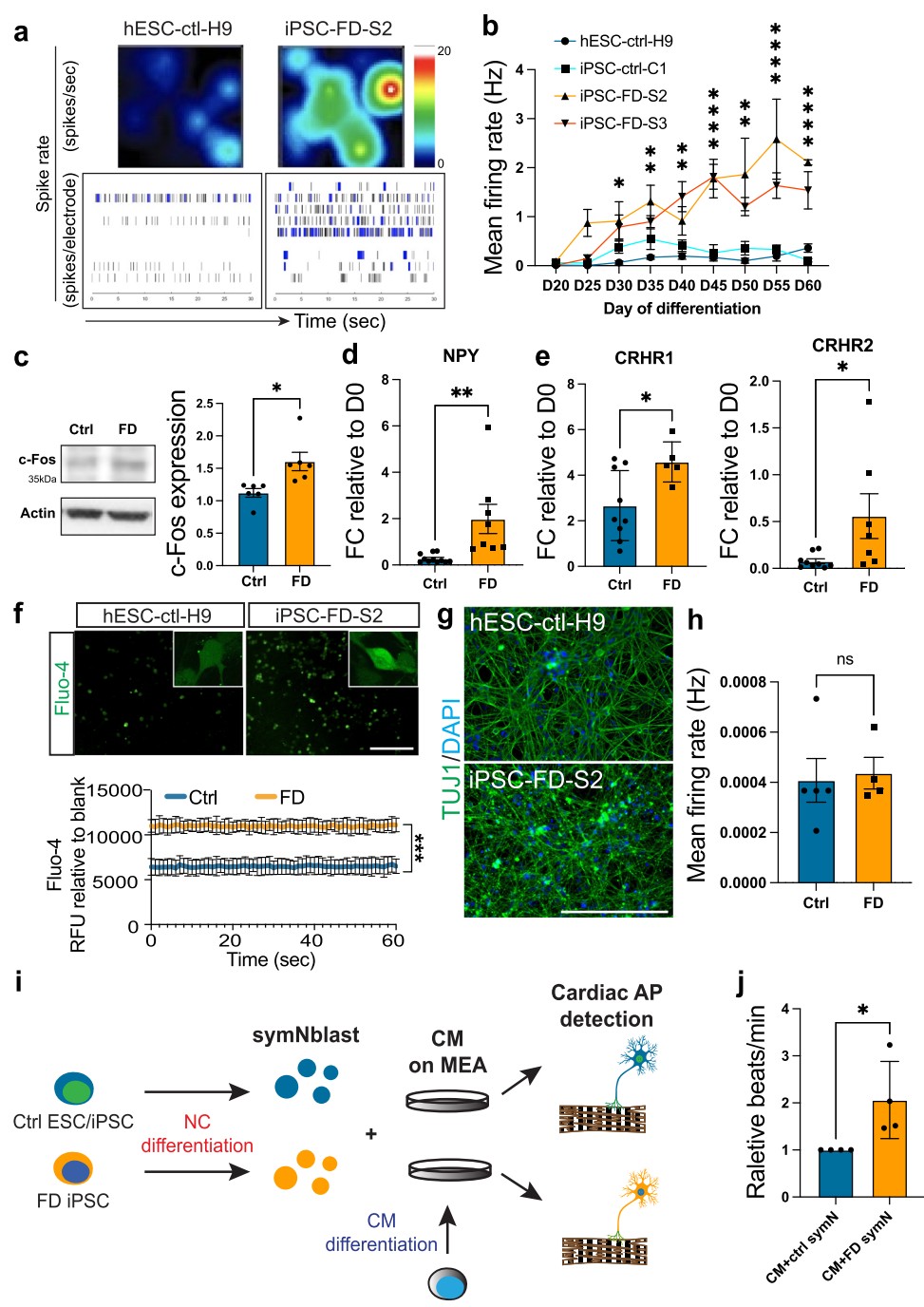

**Fig. 2 | Familial Dysautnomia sympathetic neurons (symNs) are spontaneously hyperactive. a** Multielectrode array (MEA) heatmaps of neural spike rate (top row) and burst maps (bottom row, black bars represent spikes and blue bars represent bursts) on day 35, show representative patterns of FD symN hyperactivity. **b** SymN firing frequencies are compared using MEA in a time course. $N$ = (D20-H9: 9, C1: 10, S2: 8, S3: 6; D25-H9: 9, C1: 10, S2: 6, S3: 8; D30-H9: 14, C1: 16, S2: 9, S3: 9; D35-H9: 10, C1: 10, S2: 18, S3: 12; D40-H9: 7, C1: 13, S2: 3, S3: 10; D45-H9: 3, C1: 4, S2: 3, S3: 7; D50-H9: 3, C1: 8, S2: 4, S3: 11; D55-H9: 5, C1: 6, S2: 5, S3: 4; D60-H9: 6, C1: 7, S2: 3, S3: 13) biological replicates. Two-way ANOVA followed by Šídák multiple comparisons. Control and FD were compared. Error bar represents the SEM. *$P$ < 0.05, **$p$ < 0.01, ****$P$ < 0.0001. **c** Comparison and quantification of c-Fos expression by western blot between control and FD. $N$ = 6 biological replicates. Two-tailed Student's t-test. Error bar represents the SEM. *$p$ < 0.05. RT-qPCR analysis of *NPY* (**d**) and *CRHR1/2* (**e**) levels between control and FD. For *NPY*, $N$ = (ctrl: 11, FD: 8) biological replicates. For *CRHR1*, $N$ = (ctrl: 9, FD: 5) biological replicates. For *CRHR2*, $N$ = (ctrl: 9, FD: 7) biological replicates. Two-tailed Student's t-test. Error bar represents the

SEM. *$p$ < 0.05, **$P$ < 0.01. **f** Ca²⁺ image using Fluo-4 shows higher Ca²⁺ influx in FD symNs. Images in top row show representative levels of Ca²⁺ in symNs. ELISA-based time lapse recording in bottom row shows constantly higher Ca²⁺ levels in FD symNs. $N$ = 4 biological replicates. Two-way ANOVA followed by Šídák multiple comparisons. Error bar represents the SEM. ***$p$ < 0.001. **e** CNS neurons differentiated from ctrl and FD hPSCs are Tuj1⁺ (green, **g**), and show no difference in neural activity via MEA (**h**). $n$ = (ctrl: 5, FD: 4) biological replicates. Two-tailed Student's t-test. Error bar represents the SEM. **i** Cartoon shows the rationale of symN and cardiomyocyte (CM) co-culture design. Day 14 sympathoblasts from control and FD hPSCs are plated on day 15 cardiomyocytes from hESC-ctrl-H9. **j** Quantification of CM beating rate with control or FD symN shows that FD symNs activate CMs at a higher level. $N$ = 4 biological replicates. Two-tailed Student's t-test. Error bar represents the SEM. *$p$ < 0.05. Scale bars represent 200 μm. In **b**–**f** and **j**, data from hESC-ctrl-H9 and iPSC-ctrl-C1 are pooled as control; data from iPSC-FD-S2 and iPSC-FD-S3 are pooled as FD. See also Supplementary Fig. 5. Source data are provided as a Source Data file.

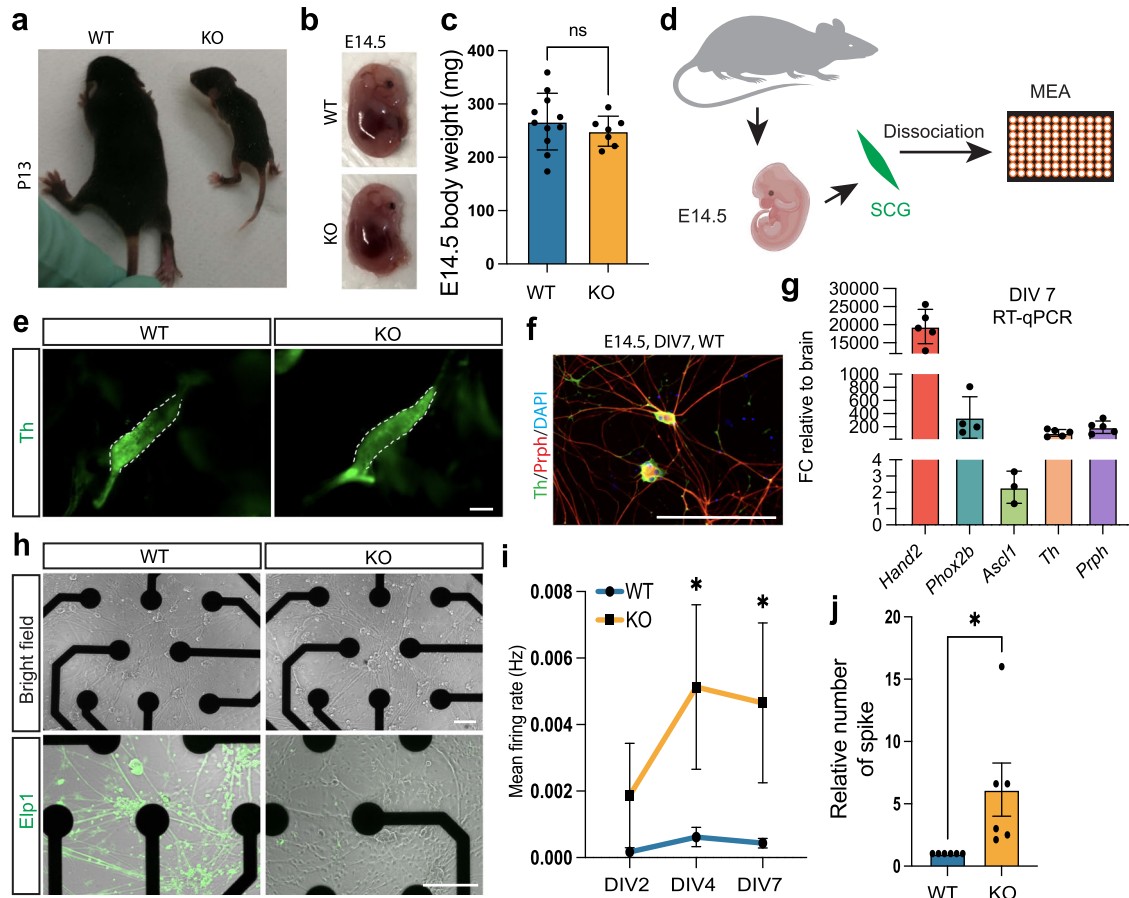

**Fig. 3 | Sympathetic neuron hyperactivity in Familial Dysautonomia mouse models. a** A survived P13 *wnt1*/Sox10-cre;*Elp1^LoxP/LoxP*-CKO mouse, has reduced body size. **b, c** E14.5 CKO embryos have no difference in appearance (**b**) as well as in body weight (**c**). *n* = (WT: 11, KO: 7) biological replicates. Two-tailed Student's *t*-test. Error bar represents the SEM. **d** Schematic illustration of superior cervical ganglia (SCG) neuron culture and experimental design. **e** Whole mount immunostaining of Th⁺ SCGs shows that E14.5 CKO SCGs have similar size and shape compared to control. **f** Immunostaining shows Th⁺ (green)/Prph⁺ (red) neurons in culture after one week. **g** RT-qPCR shows symN gene expressions on day in vitro 7 (DIV7). *N* = (*Hand2/Th/Prph*: 5, *Pho2b*: 4, *Ascl1*: 3) biological

replicates. Error bar represents the SEM. **h** Dissociated SCG neurons on MEA plates show similar neural morphology and distribution at lower magnitude (top row). Immunostaining of Elp1 (bottom row) confirms Elp1 depletion in CKO SCGs. **i** SCG neuron MEA recording in time course shows CKO symN hyperactivity. *N* = (DIV2-WT: 6, KO: 3; DIV4-WT: 7, KO: 3; DIV7-WT: 8, KO: 4) biological replicates. Two-way ANOVA followed by Šídák multiple comparisons. Error bar represents the SEM. *\*p* < 0.05. **j** Relative spike number on DIV7 is quantified by MEA. *N* = 6 biological replicates. Two-tailed Student's *t*-test. Error bar represents the SEM. *\*p* < 0.05. Scale bars represent 200 μm. See also Supplementary Fig. 6. Source data are provided as a Source Data file.

employed more FD and control iPSC lines (Supplementary Fig 7b) and assessed their electric activity in symNs (Supplementary Table 3, Supplementary Fig. 7c, d). These lines have various, but defined genotypes in *ELP1* and *LAMB4*, i.e., the severe (S) lines carry both the homozygous *ELP1* and the heterozygous *LAMB4* mutations and the mild (M) lines only carry the homozygous *ELP1* mutation and are wild type in LAMB4 (Supplementary Table 3). *LAMB4* is a candidate gene that we previously identified in three patients to act as a modifier to *ELP1* when mutated, and it is thought to induce a more severe clinical phenotype[29]. iPSC-ctrl-652 is a healthy donor line from a 11 year old female, which matches the age of our FD lines. iPSC-carrier-A1 is a phenotypically healthy FD carrier, and the parent of FD-S2. iPSC-FD-mild-M1/M2 are FD patients with mild clinical symptoms and a mild phenotype in the stem cell-based model[29]. iPSC-FD-mild-M4 is a phenotypically mild FD line that we reprogrammed here. We assessed electric activity of these additional lines and found that iPSC-ctrl-652 and iPSC-carrier-A1 are not hyperactive (Supplementary Fig. 7c), while M1, M2 and M3 all are hyperactive (Supplementary Fig. 7d and Supplementary Table 3). To explain these results, we first followed the hypothesis that hyperactivity could be a result of the *LAMB4* modifier mutation, which was previously described[29] to produce a more severe phenotype in three FD patient derived iPSC lines. While this finding

was limited in its scope of number of patients and needs to be validated for the larger FD population, here we assessed two severe FD (S2, S3, *ELP1^{-/-}*, *LAMB4^{+/-}*) and three mild FD (M1, M2, M4, *ELP1^{-/-}*, *LAMB4^{+/+}*) patient lines. We found that all 5 FD symNs were hyperactive (Fig. 2b, Supplementary Fig. 7d, and Supplementary Table 3), despite the mild lines not carrying the *LAMB4* mutation, suggesting that the *LAMB4* mutation is not resulting in symN hyperactivity. Next, we follow the hypothesis that the dosage, i.e., the level of ELP1 protein underlies hyperactivity. It has been shown in the FD mouse that severity of phenotype is a consequence of Elp1 dosage[16]. We indeed show that more differentiated neurons (i.e., symNs) have lower levels of *ELP1* splicing and are thus likely more vulnerable to its reduction compared to undifferentiated cells (Supplementary Fig. 7e). Similarly, ELP1 protein levels in iPSC-FD_rescued-T6 are intermediate, but higher than FD and significantly lower than iPSC-carrier-A1 (Supplementary Fig. 7f). Previous studies[29] have shown that ELP1 protein levels in mild FD neural crest are also higher than in severe FD lines. Thus, it is possible that the dosage of ELP1 protein contributes to symN hyperactivity. Together, this data suggests that symN hyperactivity roots in a mechanism that is directly or indirectly dependent on *ELP1*.

We next assessed whether hyperactivity in symNs may be caused by defects in the incoming signal (triggering/regulating action

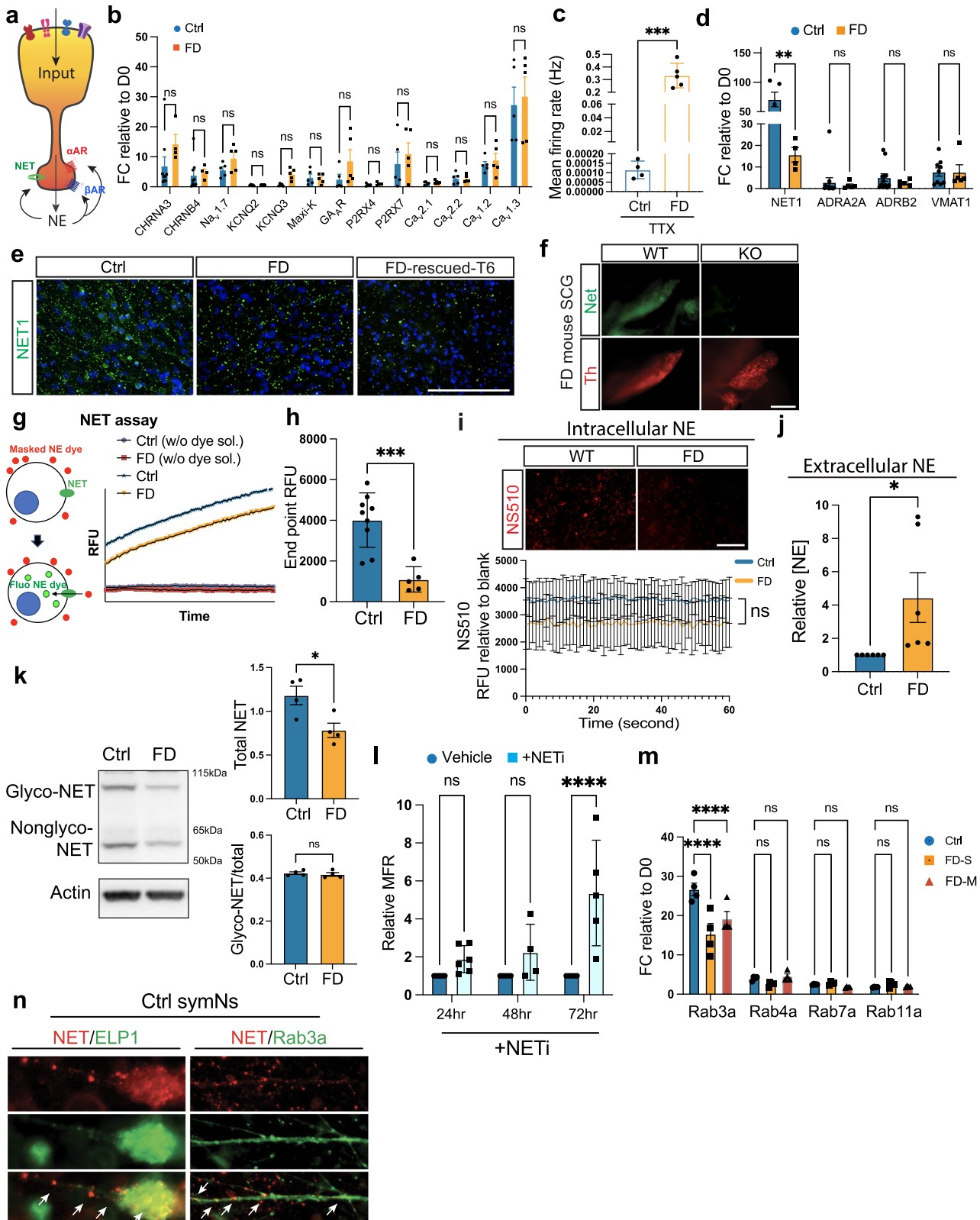

potentials) or at the outgoing signal (neurotransmitter release/re-uptake, (Fig. 4a)). To investigate the first question, we compared a selection of receptors and transporters that regulate neuronal electrophysiology and have been reported in symNs, including nicotinic receptors (CHRNs), sodium channels (Na$_v$), potassium channels (KCNs and Maxi-K), GABA receptor (GA$_A$R) and calcium channels (P2RX and Ca$_v$). We assessed differential expression thereof and found no

significant difference between control and FD (Fig. 4b). We further blocked the incoming signal with the sodium channel blocker tetro-dotoxin (TTX) in both control and FD symNs. We found no change in hyperactivity (Fig. 4c), thus confirming that it is caused by a cell-intrinsic mechanism. This result also reduces the potential concern that autapses, which have been shown to form in ex-vivo cultured symNs[60], lead to FD hyperactivity, since TTX blocks incoming signals

**Fig. 4 | Norepinephrine transporter (NET) deficiency leads to Familial Dysautonomia sympathetic neuron hyperactivity. a** Cartoon representing the rationale of the screening of ion channels/receptors/transporter that may contribute to electric activity changes in symNs. **b** Screening by RT-qPCR shows no difference in expressions of ion channels/receptors receiving presynaptic signals between control and FD. N = (CHRNA3/CHRNB4-ctrl: 9, FD: 4; other genes: 5) biological replicates. Two-tailed multiple unpaired Student's t-test. Error bar represents the SEM. **c** MEA assay still shows hyperactive FD symNs after treatment with 100 μM TTX. N = (ctrl: 4, FD: 5) biological replicates. Two-tailed Student's t-test. Error bar represents the SEM. ***P < 0.001. **d** RT-qPCR shows decreased NET RNA levels in FD symNs. N = (NET-ctrl: 5, FD: 4; ADRA2A-ctrl: 12, FD: 5; ADRB2-ctrl: 12, FD: 5; VMAT1-ctrl: 9, FD: 5) biological replicates. Two-tailed multiple unpaired Student's t-test. Error bar represents the SEM. **P < 0.01. **e** Immunostaining for NET supports decreased NET levels in FD symNs. **f** FD Elp1-CKO mouse superior cervical ganglia (SCG) have reduced NET staining. **g** Cartoon (left) illustrates the principle of NET reuptake assay. An NE simulating fluorophore is incubated with symNs in a fluorescence-masking reagent, which blocks the fluorescent signals (red dots outside of symNs). Only when the fluorophore is taken up by NET, the intracellular GFP signals can be detected (green dots inside of symNs). The chart (right) shows the trend of NE reuptake efficiency between control and FD symNs over time. **h** Quantification of the end point GFP intensities between control and FD symNs.

N = (ctrl: 9, FD: 5) biological replicates. Two-tailed Student's t-test. Error bar represents the SEM. ***P < 0.001. **i** NE imaging using NS510 shows lower intracellular NE in FD symNs. Images in top row show representative levels of NE in symNs. ELISA-based time lapse recording in bottom row shows FD symNs has lower but nonsignificant NE constantly. N = 4 biological replicates. Two-way ANOVA followed by Šídák multiple comparisons. Error bar represents the SEM. **j** ELISA assay shows higher secreted NE in FD symNs. N = 6 biological replicates. Two-tailed Student's t-test. Error bar represents the SEM. *P < 0.5. **k** Western blot analysis of NET glycosylation. N = 4 biological replicates. Two-tailed Student's t-test. Error bar represents the SEM. *P < 0.5. **l** Healthy control symNs are treated with NETi (2 μM) and electric activity is measured by MEA overtime. Neural activity gradually increases after the treatment. N = (24 h: 6, 48 h: 4, 72 h: 5) biological replicates. Two-way ANOVA followed by Šídák multiple comparisons. Error bar represents the SEM. ****P < 0.0001. **m** RT-qPCR shows reduced RAB3A in FD symNs. N = 4 biological replicates. Two-way ANOVA followed by Šídák multiple comparisons. Error bar represents the SEM. ****P < 0.0001. **n** Immunostaining shows co-localization of NET/ELP1 and NET/RAB3A. Scale bar in **n** represents 50 μm. In **b–d**, **h–m** data from hESC-ctrl-H9 and iPSC-ctrl-C1 are pooled as control; in **b–d**, **h–k**, **m** data from iPSC-FD-S2 and iPSC-FD-S3 are pooled as FD-S; in m data from iPSC-FD-M1 and iPSC-FD-M2 are pooled as FD-M. Scale bars represent 200 μm. See also Supplementary Figs. 7–9. Source data are provided as a Source Data file.

from potential self-innervation/stimulation. We next assessed outgoing signals that could trigger hyperactivity. Expression of VMAT, important for NE vesicle transport, α2AR and β2ARs, involved with NE re-uptake were not expressed differently. However, NET, which reuptakes nearly 90% of secreted NE, and thus plays a critical role in extracellular NE clearance was expressed significantly lower in FD symN cell bodies and axons (Fig. 4d). NET deficiency has been reported in multiple neural disorders, in which SNS dysregulation is involved[61,62]. We confirmed reduced NET protein in FD symNs by IF, as well as in symNs from iPSC-FD_rescued-T6 in vitro (Fig. 4e) and in the FD mouse sympathetic ganglia (Fig. 4f). We further showed reduced uptake of fluorescently labeled, synthetic NE by NET in FD symNs, using a NET reuptake assay (Fig. 4g, h). Less NETs on the surface of FD symNs is expected to lead to reduced NE levels inside the cell and excessive NE levels in the extracellular space in FD. Indeed, we showed diminished (though not significantly) NE levels inside FD symNs (Fig. 4i), using a novel NE tracker NS510, which is a turn-on probe that allows live cell imaging of intracellular NE dynamics[63]. Accordingly, ELISA measurements of NE in symN media revealed that in FD symNs more NE is present in the extracellular space (Fig. 4j). For the NET protein to be trafficked to the cell surface and thus become functional, it has to be phosphorylated and glycosylated[64]. Thus, we assessed those modifications on NET in FD and control symNs. We did not detect a significant difference in glycosylation (Fig. 4k), indicating that NET function may not be affected beyond the lower expression levels. To test if NET inhibition is sufficient to induce the hyperactive phenotype, we treated healthy control symNs with nomifensine, a norepinephrine-dopamine reuptake inhibitor (NETi) for 24–72 h to allow the accumulation of extracellular NE and recorded neural activities by MEA. In this analysis, control symNs gradually become hyperactive (Fig. 4l), indicating that indeed NET deficiency in FD symNs may underlie intrinsic hyperactivity.

We next aimed at understanding the potential connection between the FD ELP1 mutation, its consequences, and NET reduction, all leading to symN hyperactivity. Previous studies on FD PNS neurons differentiated from a FD hESC line has revealed an important vesicular transportation role of ELP1[65]. In addition, a recent study using FD mouse models has shown enriched cytosolic Elp1 expression in symNs and defective NGF retrograde transport in FD symNs due to decreased Elp1 levels[20]. We assessed these findings in our human FD symNs. We replated day 14 sympathoblasts on fluidic compartmental culture devices with an NGF gradient in the axon chambers (Supplementary Fig. 8a). Similar to previous results, FD symNs showed impaired axon outgrowth, likely due to defects in NGF retrograde transport

(Supplementary Fig. 8b), supporting the notion of the importance and possible defect in vesicular transportation in FD symNs. We further searched the literature and found the importance of RAB proteins for vesicle trafficking, a group of small GTPase that belongs to the RAS superfamily, and its connection to ELP1. RAB3A and ELP1 colocalization was found decreased in PNS neurons[65]. It was shown that in yeast ELP1 activates Rab to regulate exocytosis[66]. ELP1 was also found colocalized with RAB7-associated late endosomes[67,68], which is involved in the pathogenesis of multiple HSAN subtypes[69]. RAB proteins are also important for catecholamine regulation. For instance, RAB4 is crucial for dopamine transporter (DAT) and serotonin transporter (SERT) membrane trafficking[70,71]. RAB11 has been identified to participate in NET internalizing pathways[72]. Importantly, Goffena et al., found that proteins of a specific length and AA:AG codon ratio were most affected by elongator dysfunction in FD mouse sensory neurons, and many RAB proteins fall into this category[4]. We, therefore, hypothesized that ELP1 may affect symN activity by regulating NET membrane trafficking via RAB proteins. We first compared the levels of a panel of RAB proteins in control and FD symNs. We found that RAB3 expression was significantly decreased in FD symNs, both in the severe FD as well as the mild FD lines (Fig. 4m). Furthermore, in control symNs NET and ELP1 colocalize, as well as RAB3 and NET (Fig. 4n), suggesting that RAB3 is involved in ELP1-mediated trafficking of NET that contributes to symN hyperactivity. We then conducted bulk RNA sequencing of healthy control (hESC-ctrl-H9, N = 4) and FD (iPSC-FD-S2, N = 4) symNs at day 35 (Supplementary Fig. 9a–c). We found that LAMB4 was not differentially expressed supporting the notion that LAMB4 does not play a critical role in symN hyperactivity. However, we found that many (over 100) RAB proteins were up or downregulated in FD symNs compared to control (Supplementary Fig. 9d), supporting our findings of their importance in FD symN hyperactivity. In sum, these results indicate that symNs in FD are intrinsically hyperactive due to defects in autoregulation of their neurotransmitter. That ELP1 connects to NET defects via RAB proteins and defective trafficking and that leads to increased activation of symN target tissue.

## Mini drug screen reveals potential treatments of hyperactive FD symNs

With these exciting findings, we wanted to test whether this hPSC model system could be used as a platform to test drugs and therefore as a future drug discovery tool. To do so, we selected several drugs whose therapeutic potential have been studied at the clinical or experimental level in FD or other PNS diseases. We tested those drugs on day 35 symNs, the timepoint when hyperactivity in FD symNs first

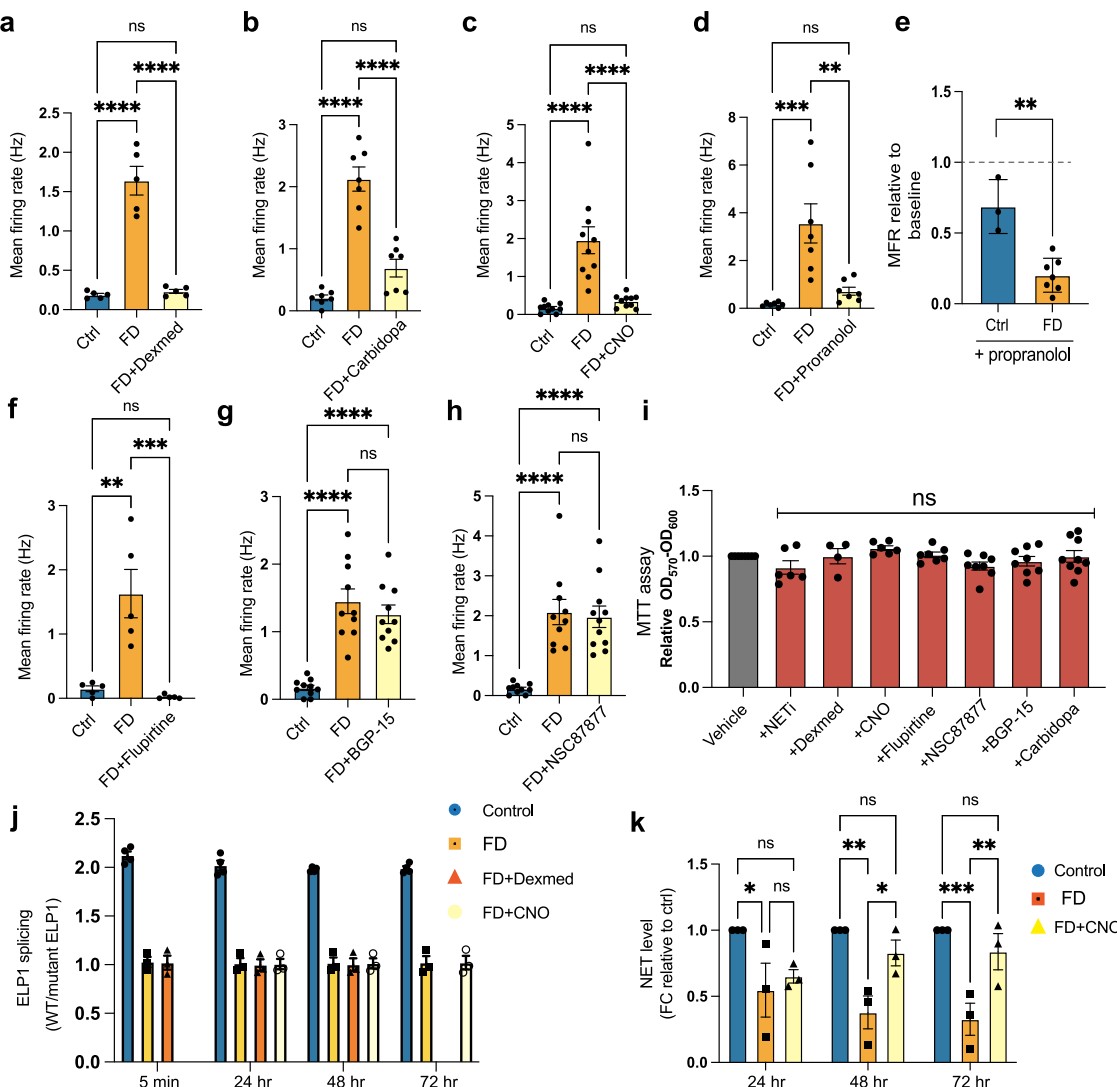

**Fig. 5 | Mini drug screen to rescue sympathetic neuron hyperactivity in Familial Dysautonomia.** A selection of drugs are added to FD symN on day 35, followed by MEA measurements. The dosage and treating conditions of each are: **a** Dexmed (60 μM) for 5 min. $N = 5$ biological replicates. One-way ANOVA followed by Tukey's multiple comparisons. Error bar represents the SEM. ****$P < 0.0001$. **b** Carbidopa (25 μM) for 5 min. $N = 7$ biological replicates. One-way ANOVA followed by Tukey's multiple comparisons. Error bar represents the SEM. ****$P < 0.0001$. **c** Clozapine (CNO, 3 μM) for up to 72 h in order to stimulate NET expression. $N = 10$ biological replicates. One-way ANOVA followed by Tukey's multiple comparisons. Error bar represents the SEM. ****$P < 0.0001$. **d** Propranolol (1 μM) for 5 min. $N = 7$ biological replicates. One-way ANOVA followed by Tukey's multiple comparisons. Error bar represents the SEM. **$P < 0.01$, ***$P < 0.001$. **e** Shows the differing ratio of firing after propranolol treatment of both ctrl and FD neurons. $N = $ (ctrl: 3, FD: 7) biological replicates. Two-tailed Student's t-test. Error bar represents the SEM. **$P < 0.01$. **f** Flupirtine (15 μM) for 5 min. $N = 5$ biological replicates. One-way ANOVA followed by Tukey's multiple comparisons. Error bar represents the SEM. **$P < 0.01$, ***$P < 0.001$. **g** BGP-15 (10 μM) for 5 min–72 h. $N = 10$ biological replicates. One-way

ANOVA followed by Tukey's multiple comparisons. Error bar represents the SEM. **h** NSC87877 (10 μM) for 5 min–72 h. $N = $ (ctrl/FD: 10, FD + NSC: 11) biological replicates. One-way ANOVA followed by Tukey's multiple comparisons. Error bar represents the SEM. The results are compared with those in untreated control and FD symNs. **i** The toxicity of all drugs used in this study (including NETi) is evaluated by MTT assay. $N = $ (NETi:6, Dexmed: 4, CNO: 6, Flupirtine: 7, NSC87877: 8, BGP-15: 8, Carbidopa: 9; each was normalized to its own vehicle control with same number) biological replicates. One-way ANOVA followed by Tukey's multiple comparisons. Error bar represents the SEM. **j** *ELP1* splicing was compared using RT-qPCR after the treatments of Dexmed (5 min–48 h) and CNO (24–72 h). $N = $ (ctrl: 4, rest: 3) biological replicates. Error bar represents the SEM. **k** NET level was compared using RT-qPCR after the treatment of CNO. $N = 3$ biological replicates. Two-way ANOVA followed by Šídák multiple comparisons. Error bar represents the SEM. *$P < 0.05$. **$P < 0.01$. ***$P < 0.001$. In **a–d** and **f–h**, data from hESC-ctrl-H9 and iPSC-ctrl-C1 are pooled as control; in **a–h**, data from iPSC-FD-S2 and iPSC-FD-S3 are pooled as FD. In **j** and **k**, hESC-ctrl-H9 and iPSC-FD-S2 are used as control and FD, respectively. See also Supplementary Fig. 10. Source data are provided as a Source Data file.

peaks, yet neurodegeneration has not occurred. Dexmedetomidine (Dexmed) is a novel selective α2-AR enhancer, shown to relieve dysautonomic crisis symptoms in some FD patients[73]. Carbidopa is a selective AAAD inhibitor that blocks NE and DA synthesis; its effect of rescuing failed baroreflex functions were shown in some FD patients[74]. We found that indeed, both drugs reduce the exaggerated firing activity of FD symNs in our model (Fig. 5a, b and Supplementary Fig. 10a), supporting the notion that our hPSC-based model is useful for drug testing. We next sought to treat FD symNs by targeting the

NET pathway. Clozapine (CNO) is an atypical antipsychotic medicine, which has been demonstrated to elevate NET expression in chromaffin cells[75]. Again, clozapine reduces FD symN hyperactivity significantly in our model (Fig. 5c and Supplementary Fig. 10b). Lastly, propranolol, a β2AR inhibitor reduces FD hyperactivity as well (Fig. 5d). We compared the responses of control and FD symNs after propranolol treatment at the same dosage and found that propranolol inhibits FD symN activity even more compared to its action in control neurons (Fig. 5e). This may imply that β2AR signaling in FD symNs is hypersensitized, further

driving the loop of hyperactivity. Together, these results highlight the usefulness of our modeling platform to assess drugs for FD patients, and together with our NETi result on control symNs (Fig. 4l), we show that manipulating NET function is an effective strategy for modulating symN activity.

Next, we aimed to test drugs that have been used in PNS-related neural disorders, but not in FD, to our knowledge. Previous research in ALS, a neurodegenerative disease with motor neuron denervation, revealed motor neuron hyperactivity due to potassium channel mal-function and identified flupiritine as a modulator[21]. We tested this drug on FD symNs, and indeed found that flupiritine decreased FD hyper-activity (Fig. 5f). Finally, we examined NSC87877 and BGP-15, two drugs that have been studied to prevent PNS neuron degeneration[18,20]. NSC87877 is a tyrosine phosphatase inhibitor shown to prevent FD symN death from impaired retrograde signaling in FD mice[20]. BGP-15 is a small hydroxylamine compound that improves mitochondrial function[18]. We found that treatment of FD symNs with NSC87877 or BGP-15 does not rescue hyperactivity (Fig. 5g, h and Supplementary Fig. 10c, d), likely due to their mechanism of action that targets neu-ronal survival rather than activity. All drugs used in this study do not affect symN survival (Fig. 5i).

Although the drugs we tested here are unlikely to rescue *ELP1* splicing based on previous mechanistic studies (most of the drugs target ion channels or signaling receptors to interfere with signal transductions), we verified *ELP1* splicing levels after treatments of symNs with Dexmed and CNO, which represent a typical FD medica-tion and a newly discovered drug with a hypothetical mechanism, respectively. Both drugs did not alter wild type *ELP1* splicing levels (Fig. 5j). However, CNO rescued NET expression (Fig. 5k) as expected, suggesting it to be a potentially beneficial drug compound for FD patients.

Our symN platform and FD model are instrumental for testing current drugs and thus promising for future drug discovery attempts. Furthermore, the results from these drug treatments further strengthen our conclusion about the molecular mechanism underlying symN hyperactivity. In healthy symNs (Fig. 6, top) NE is released into the extracellular space and about 10% of it is bound by the target tissue. Another 10% is taken up by $\alpha_2$AR on the neuron itself, which signals to inhibit further NE release and 10% is taken up by the $\beta_2$AR, which signals to activate NE release. The final 70% of NE is taken up via NET. RAB proteins are important for NET trafficking to the cell mem-brane (Fig. 4m, n). FD symNs have less NET molecules available (Fig. 6, bottom), due to reduced expression as well as RAB-mediated mem-brane trafficking problems. Thus, NE is depleted inside the cells and in excess outside the cells. That excess can bind the target tissue as well as to both $\alpha_2$-AR and $\beta_2$AR, triggering FD symN hyperactivity.

Together, we show that our hPSC-derived symN platform is a powerful tool to select potential drug compounds for future treatment options of SNS hyperactivity in FD.

## Discussion

Few in vitro differentiation protocols have been published to date for the generation of postganglionic symNs from hPSCs[31–34]. Our protocol has the following advantages, which makes it ideal for our FD disease modeling application: 1. It is highly efficient and produces a relatively pure symN population, i.e., >90% NCCs[30] and >75% neurons, of which >90% are peripheral symNs (Supplementary Figs. 1, 2). 2. It generates functional and mature symNs and is practical, as we have developed a cryopreservation option (Supplementary Fig. 1i, j). 3. Our protocol recapitulates the proper developmental steps of symN development and was the pioneering protocol to be successfully employed for dis-ease modeling in Zeltner et al.[29], and here.

FD is a developmental disorder with symptom onset at birth[5]. Accordingly, we show reduced generation of NCCs from FD iPSCs (Fig. 1b–e) and reduced cell numbers and lower cell survival at the

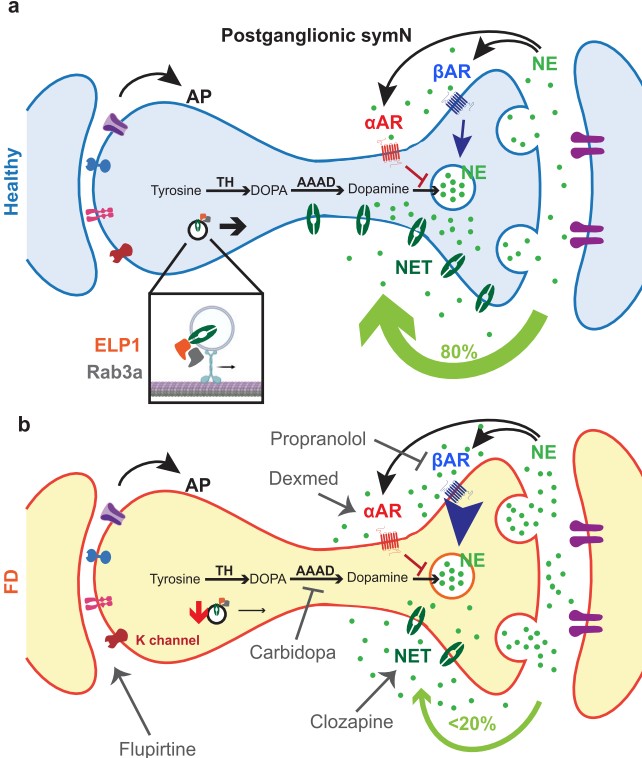

**Fig. 6 | Model.** In healthy postganglionic sympathetic neurons (symNs) (**a**), nor-epinephrine (NE) secretion in the axon terminal is properly regulated by nor-epinephrine transporter (NET), which reuptakes about 80% of released NE, α adrenergic receptors (αAR), which downregulate NE secretion and β adrenergic receptors (βAR), which upregulate NE secretion. NET is trafficked to the cell membrane with the help of RAB proteins. In FD symNs (**b**), NET expression is decreased, leading to oversecreted NE. NET trafficking might be affected by the downregulation of RAB3A. In addition, our results indicate that βAR in FD symNs might have stronger response than healthy symNs, which might further strengthen the release of NE and symN activity. Several therapeutic targets are identified in this study using a selection of clinical compounds: flupirtine activates potassium channels (K channel) and induces neural depolarization; carbidopa blocks NE synthetic pathway and thus reduces NE overspill; dexmedetomidine (Dexmed) activates αAR while propranolol inhibits βAR. All these drugs suppress FD symN hyperactivity in vitro.

sympathoblast stage (Fig. 1f–h). However, the symNs that passed through early development, do mature comparably to control neurons (Fig. 1i–k). These observations correlate with FD patient autopsies that showed reduced volume in the sympathetic ganglia[13]. FD animal data from conditional knock-out (CKO) and other mouse models[19] are slightly different, in that NCC development ensues normally[14,44], but neuron numbers are decreased due to maldevelopment[14,44], defective innervation[15,20], reduced neuron survival[14,20] and abnormal NGF retro-grade transport[20,76,77]. However, the end-result of overall reduced symNs is consistent with FD mouse models.

Using our in vitro disease model for FD, we found that FD symNs are intrinsically hyperactive. We confirmed this finding via electro-physiology, associated gene expression, Ca²⁺-imaging and in co-cultures with target tissue, where cardiomyocytes beat faster when coupled with FD symNs (Fig. 2). We showed that symN hyperactivity is conserved in two CKO FD (*elp1*^LoxP/LoxP^) mouse models, driven by either *WNT1*-cre or *SOX10*-cre (Fig. 3 and Supplementary Fig. 6). Neural hyperactivity has been shown in a variety of neurological disorders and is associated with neurodegeneration. Others have reported the causality of motor neuron hyperactivity and neurode-generation in ALS using patient-iPSC model[21]. It has been suggested that network activity may lead to neurodegeneration in the brain of

Alzheimer's disease (AD)[78,79], and an hPSC-based AD model that recapitulates neural hyperactivity was proposed recently by the Lipton group[22]. We indeed found that symNs from FD animals, that are hyperactive, show signs of degeneration quickly in culture even in the presence of NGF (60 ng/ml) (Supplementary Fig. 6c). Future investigations will show if pharmacological prevention of hyperactivity may prevent neurodegeneration in FD.

The SNS is an important component of the stress response system and is crucial for regulation of body homeostasis. FD patients have difficulties regulating arousal, and stress can trigger dysautonomic crisis[6]. Could it be possible then that FD symNs are more susceptible to stress signaling from the brain? During a stressful state, the extra-hypothalamic areas and hypothalamus in the brain release corticotropin-releasing factor (CRF), a stress-related neuropeptide, which has been found to regulate the SNS as well as symN activity[47]. We did find elevated expression of CRF receptors (*CRHR1/2*) in FD symNs. NPY is another important neuropeptide involved in stress modulation both in the CNS and PNS. In PNS, NPY is mainly produced by symNs, its levels have been positively correlated to symN activity, and NPY released by symNs has a vasoconstriction effect[46,80]. Elevated plasma NPY levels are also found in stress-related disorders, such as hypertension and heart failure[46]. Indeed, we detected elevated NPY levels in FD symNs (Fig. 2d), supporting the hyperactivity phenotype in FD symNs, but also suggesting that FD SNS may be more vulnerable to stress stimulation.

We assessed if the FD causative point mutation in *ELP1* that 99.5% of FD patients carry homozygously is the cause of symNs hyperactivity. Interestingly, heterozygous rescue of the *ELP1* mutation did not rescue the hyperactivity phenotype (Supplementary Fig. 7a), even though an also ELP1-heterozygous FD carrier line maintained physiological levels of electric activity (Supplementary Fig. 7c). To understand this puzzling result, we first followed the hypothesis that the FD modifier mutation in *LAMB4*[29], which is present in the FD iPSC lines used so far might be at fault. However, symNs derived from three mild FD iPSC lines (all with the genotype $ELP1^{-/-}$, $LAMB4^{+/+}$) all remained hyperactive (Supplementary Fig. 7d and Supplementary Table 3), indicating that *LAMB4* is not the culprit. The influence of the dosage of ELP1 protein on disease severity has been shown previously[16,29]. Accordingly, we found that symNs from the hyperactive iPSC-FDrescue-T6 (Supplementary Fig. 7f) as well as the hyperactive iPSC-FD-M1/2/3[29] had lower ELP1 protein levels compared to the iPSC-carrier-A1 symNs, indicating that indeed ELP1 protein dosage might contribute to symN hyperactivity. This likely reflects variability within individuals. Nevertheless, to fully assess the effect of ELP1 rescue on symN activity, generating a homogeneous ELP1 rescue line is an important future task.

We hypothesize that additional intrinsic defects within FD symNs are the drivers that lead to hyperactivity, which might further trigger other FD phenotypes and symptoms, including dysautonomic crisis and neurodegeneration. Literature from other SNS related disorders supports this notion. For instance, symN neurotransmitter switching from NE to epinephrine in prehypertensive rats[81] indicates that symN abnormalities prior to the onset of cardiovascular symptoms may sensitize the SNS and make it more vulnerable to disease-related stimuli. Another study showed that hypertensive symNs were able to change the metabolism of healthy cardiomyocytes to induce a hypertensive state. Vice versa, when hypertensive cardiomyocytes were co-cultured with healthy symNs, the defective response was suppressed[82].

Thus, we looked deeper into the molecular mechanism underlying FD symN hyperactivity. We showed that NET expression is reduced in FD, which leads to an accumulation of NE in the extracellular space and to diminished NE inside the cells compared to control (Fig. 4). NET deficiency has been linked to several SNS-related cardiovascular syndromes, such as orthostatic intolerance (OI), postural tachycardia syndrome (POTS)[61,83], and stress-induced

cardiomyopathy[62], both in patients and animal models[62,84]. Furthermore, a study in patients with hypertension revealed that NE reuptake function was impaired, while the baroreflex control remained unaffected[85]. We also show that NET inhibition is sufficient to trigger symN hyperactivity in healthy hPSCs (Fig. 4l). We further found that NET protein in FD is functional, i.e., glycosylation is not affected (Fig. 4k), thus the issue might be expression of the protein itself. NET is being trafficked to the cellular membrane in vesicles This process is heavily supported by the family of RAB proteins. Several findings point towards a connection of RABs and ELP1, including their co-localization[65,67,68], *ELP1*'s activation of RABs to regulate exocytosis[66] and its involvement in the pathogenesis of multiple HSAN subtypes[69]. RAB proteins are also important for catecholamine regulation, i.e., RAB4 is crucial for dopamine transporter (DAT) and serotonin transporter (SERT) membrane trafficking[70,71] and RAB11 has been identified to participate in NET internalizing pathways[72]. Most importantly, Goffena et al., found that proteins of a specific length and AA:AG codon ratio were most affected by elongator dysfunction in FD mouse sensory neurons, and many RAB proteins fall into this category[4]. Accordingly, we found RAB3A to be downregulated in all hyperactive FD symNs (Fig. 4m) and co-localized with NET and ELP1 (Fig. 4n).

As a consequence of defects in NET, in FD excessive NE accumulates in the extracellular space. This, extra NE is bound by $a_2AR$, which inhibits more NE release and $b_2AR$, which further activates more NE release[81,86–88]. Theoretically, these effects might cancel each other. However, we found that the reduction in hyperactivity after treatment with propranolol, a $b_2AR$ inhibitor is more pronounced in FD compared to propranolol treated control symNs (Fig. 5e). Thus, in FD the $b_2AR$ might be hypersensitized, which might maintain a feed forward loop, which drives hyperactivity in FD symNs. So, in a FD patient, early embryonic hyperactivity of symNs may explain later degeneration of the cells as well as hypersensitization of $b_2AR$ receptors, that together might be the important players for dysautonomic crisis. We believe that our findings are crucial to understand the FD phenotype better and should help drug development for symptomatic therapy for FD patients. Until now, the explanation for the excessive sympathetic activity in FD patients was a lack of afferent baroreflex restrain. Intrinsic neuronal sympathetic hyperactivity and reduced NET expression add a robust second mechanism that further explains the phenotype. In our in vitro system, it remains to be shown when measurable degeneration signs begin and if they can be pharmacologically countered, which is a future goal of our work. In patients, indirect measurements of symN activity at rest, i.e., not in crisis, have not shown symN hyperactivity[43]. This discrepancy might have several explanations: (i) Such measurements are indirect, including skin conductance or heart rate assessments[25] as it is not feasible to directly measure symNs in a patient. Thus, at rest symN hyperactivity might not be strong enough to translate to the downstream target. (ii) In our in vitro, miniature model system, symNs are isolated and thus the released NE is bathing the entire neuron, which provides more opportunity to activate its firing. In comparison, in vivo the excess NE would mainly affect the distal axonal region and thus require more activation, such as during stress-triggered crisis to start hyperactivity in a crisis. (iii) In vivo, in a whole organism, the counteracting parasympathetic nervous system is operating as well. Finally, a defect in the NET transporter also explains the puzzling finding of lack of fluorodopamine uptake by sympathetic post ganglionic neurons innervating the myocardium in patients with FD[89], which was not in line with clinical findings. It had previously been interpreted as a result of sympathetic denervation; however, with our findings it is likely the result of impaired NET transporter activity.

Finally, we aimed at testing if our symN disease modeling platform is useful for drug screening and drug testing. We found

that in this platform, drugs currently given to FD patients for dysautonomic crisis, i.e., dexmedetomidine[73] and carbidopa[74] were able to reduce hyperactivity (Fig. 5). Additionally, clozapine and flupiritine also could reduce hyperactivity. We further showed that neither DexMed, nor CNO modulate *ELP1* splicing, however CNO as expected increases the expression of NET in FD symNs (Fig. 5k). In contrast, NSC87877 and BGP-15 do not affect FD symN activity in our model. This might be due to the short-term treatment used here and the fact that these drugs have been shown to prevent neurodegeneration. In the future, we aim to establish neurodegeneration in our system and to test if these drugs may prevent it here as well. Together, our drug testing results support our model of the mechanism of defective NE trafficking and NET reuptake, through which hyperactivity in FD is generated and maintained (see model in Fig. 6). These results also indicate that this platform will be useful for drug screening approaches to identify and test compounds that could treat FD or other SNS-related disorders.

## Methods

### Ethics statement for human pluripotent stem cells
This study employed human embryonic stem cell lines (WA09), the use of which was approved to the Zeltner lab by WiCell. All iPSCs employed in this work were reprogrammed from human samples obtained through the public repository Coriell Research Institute.

### Stem cell cultures
hPSC lines: hES-ctrl-H9=WiCell WA09; iPSC-ctrl-C1, iPSC-FD-S2, iPSC-FD-S3, iPSC-rescued-T6, iPSC-carrier-A1, iPSC-FD-M1, iPSC-FD-M2 were characterized in Zeltner, 2016[29]. iPSC-ctrl-652 were reprogrammed from fibroblasts from a 11-year old, female, healthy donor, Coriell #GM01652, iPSC-FD-M4 were reprogrammed from fibroblasts from a 2-year-old female patient with FD, Coriell #GM04663, see table 3. Cells were maintained in feeder-free and chemically defined Essential 8 medium (Gibco, A15170-01) on 5 μg/ml vitronectin coated (Thermo Fisher/Life Technologies, A14700) cell culture plates, according to the manufacturer's instructions.

### In vitro differentiations
**Sympathetic neuron differentiation.** The differentiation protocol is modified from our previous publications[29,30]. hPSCs on day 0 were dissociated by EDTA. After dissociation, cells were mixed in day 0–1 medium that contains Essential 6 medium (Gibco, A15165-01), 0.4 ng/ml BMP4 (PeproTech, 314-BP), 10 μM SB431542 (R&D Systems, 1614) and 300 nM CHIR99021 (R&D Systems, 4423), and plated on Geltrex (Invitrogen, A1413202) coated cell culture plates at $1.25 \times 10^5$/cm² density. On day 2, change the culture medium to day 2–10 medium, that contains Essential 6 medium, 10 μM SB431542 and 0.75 μM CHIR99021. Medium is changed every other day. On day 10, NCCs were dissociated by accutase (Coring, AT104500) and resuspended on ultra-low attachment plates (Corning, 07 200 601 and 07 200 602) in day 10–14 medium that contains Neurobasal medium (Gibco, 21103-049), B27 (Gibco, 17502-048), L-Glutamine (Thermo Fisher/Gibco, 25030-081), 3 μM CHIR99021 and 10 ng/ml FGF2 (R&D Systems, 233-FB/CF). On day 14, sympathetic neuroblast spheroids were spun down and dissociated by accutase. Cells were plated on PO (Sigma, P3655)/LM (R&D Systems, 3400-010-01)/FN (VWR/Corning, 47743-654) coated plates at $1 \times 10^5$/cm² density in symN medium that contains Neurobasal medium, B27, L-Glutamine, 25 ng/ml GDNF (PeproTech, 450), 25 ng/ml BDNF (R&D Systems, 248-BD), 25 ng/ml NGF (PeproTech, 450-01), 200 μM ascorbic acid (Sigma, A8960), 0.2 mM dbcAMP (Sigma, D0627) and 0.125 μM retinoic acid (Sigma, R2625, add freshly every feeding). Medium is changed every three days until day 20, and every week after that. To eliminate contaminating cells, 0.5–1 μM aphidicolin (Cayman, 14007) can be added to the symN medium from day 20–30.

**Note.** BMP4 quality is highly lot dependent. It is highly recommended to perform titration test and decide the best BMP4 concentration for each batch of BMP4.

**CNS neuron differentiation.** The differentiation protocol is modified from our previous publications[50,90]. hPSCs on day −1 were dissociated by EDTA. After dissociation, cells were mixed in Essential 8 medium, supplied with 10 μM Y-27632 (R&D Systems, 1254) and plated on Matrigel (Corning, 1:20)-coated plates at $2.5 \times 10^5$/cm² density. On day 0, medium is switched to day 0–2 medium, that contains Essential 6 medium, 10 μM SB, 200 nM LDN193189 (Selleck Chemicals, S2618) and 2 μM XAV939 (TOCRIS, 3748). On day 3, feed with day 3–5 medium that contains Essential 6 medium, 10 μM SB and 200 nM LDN193189. On day 5–9 feed with medium that contains Essential 6 medium and 10 μM SB. On day 10, medium was changed to day 10–20 medium that contains Neurobasal medium, B27 (1:1000) and GlutaMAX (Gibco, 35050061). On day 20, neural progenitors were dissociated by accutase and resuspended in day 20–30 medium that contains Neurobasal medium, B27 (1:50), GlutaMAX and 40 μM DAPT (R&D Systems, 2634). From day 30, DAPT is removed from the medium, and 25 ng/ml GDNF, 25 ng/ml BDNF, 200 μM ascorbic acid, 0.2 mM dbcAMP is added to the the final medium.

**Cardiomyocyte differentiation.** The differentiation protocol is modified from previous publications[51,52]. hPSCs on day −2 were dissociated shortly by EDTA and split at a 1:5 ratio on plates coated with 1:20 diluted Matrigel, in Essential 8 medium supplied with 10 μM Y-27632. Prepare CDBM base medium that contains DMEM/F12 (Gibco, 11320033), 64 mg/L ascorbic acid, 13.6 μg/L sodium selenium (Sigma, S5261), 10 μg/ml transferrin (Sigma, T3309) and Chemically Defined Lipid Concentrate (Gibco, 11905031). Cells were fed with Essential 8 medium until day 0. On day 0, cells were fed with day 0 medium that contains CDBM base and 5 μM CHIR99021. Cells were fed daily thereafter. Day 1, 5, and 6 medium contains CDBM base and 0.6 U/ml heparin (STEMCELL Technologies, 07980). Day 2, 3, and 4 medium contains CDBM base, 0.6 U/ml heparin and 3 μM XAV. On day 7, CMs were dissociated by accutase and resuspended in final CM medium that contains CDBM base and 20 μg/ml insulin (Sigma, I-034) at $2 \times 10^5$ cells/cm² density on Matrigel coated plates.

**SymN and CM co-culture.** Day 14 symN progenitors were added on top of day 15 CMs maintained on Matrigel coated MEA plates at $1 \times 10^5$ cells/cm² density in symN medium plus CM medium at a 1:1 ratio. At the end of co-culture, cardiac action potential was measured by Maestro Pro multiwell plate reader (Axion BioSystems) under cardiac detection mode according to manufacturer's instructions (see below for more information of MEA measurements). Beating rate was calculated by taking the reciprocal of beating period.

**Sensory neuron differentiation.** Mechanoreceptor-enriched differentiation was performed as previously described[91,92]. Prior to differentiation, plates were coated with vitronectin at 1:100 dilution in PBS and stored at 37 °C o/n. The next day (day 0), hPSCs were harvested using EDTA for 15 min, resuspended in PBS, centrifuged at RT for 5 min at $200 \times g$, and resuspended in Essential 6 Medium (E6) containing 10 μM SB431542, 1 ng/mL BMP4, 300 nM CHIR99021, and 10 μM Y-27632. Cells were plated at 200,000 cells/cm2 density on the vitronectin plates. The following day, the cells were fed with the same medium. On day 2, cells were fed with D2-12 medium, which is E6 + 10 μM SB431542, 0.75 μM CHIR99021, 2.5 μM SU5402 (Biogems, 2159233), and 2.5 μM DAPT. Cells were fed every 48 h between day 2 and day 6.

On day 6, cells were dissociated with Accutase for 20 min, washed with PBS, and resuspended in SN medium: neurobasal medium containing N2, B-27, 2 mM L-glutamine, 20 ng/mL GDNF, 20 ng/mL BDNF,

25 ng/mL NGF, 600 ng/mL laminin-1 and fibronectin, 1 μM DAPT, and 0.125 μM retinoic acid. Cells were replated at a density of 250,000 cells/cm2 onto plates previously coated with 15 μg/mL poly-L-ornithine hydrobromide, 2 μg/mL mouse laminin-1, and 2 μg/mL human fibronectin (PO/LM/FN). Cells were fed every 2–3 until they were fixed.

## Compartmental culture of hPSC-symNs using microfluidic devices

Day 14 sympathetic neuroblasts were plated to PO/LM/FN coated OMEGA4 Neuronal Co-Culture Device (eNUVIO) at $1 \times 10^5$/cm² density in symN medium on one side of the culture chambers (this is defined as the cell chamber). SymN medium contains NGF was fed to both the cell chamber and the empty chamber (this is defined as the axon chamber) to allow symN differentiation. On day 20, NGF containing medium was fed only to the axon chamber, while cell chamber was fed with symN medium without NGF. The medium volume of the axon chamber should be always two-fold higher than the cell chamber to create NGF gradient. Axon outgrowth on the axon chamber was assessed on day 30.

## SCG neuron purification and culture

*SOX10*-cre;Elp1^Loxp/LoxP CKO C57BL/6 mice were provided by Dr. Hong-Xiang Liu's lab at University of Georgia and *wnt1*-cre;Elp1^Loxp/LoxP CKO C57BL/6 mice by Dr. Frances Lefcort's lab at Montana State University. The use of animals was approved by the Institutional Animal Care and Use Committee at the University of Georgia and Montana State University. The study was performed in compliance with the National Institutes of Health Guidelines for the care and use of animals in research. The sex of embryos was not tested or assessed in this study. 6 litters of *SOX10*-cre;Elp1^Loxp/LoxP CKO mice and 3 litters of *wnt1*-cre;Elp1^Loxp/LoxP CKO mice were used. Each litter has about 8–12 pups with the chance of <25% to be KO. Other siblings mostly WT, but also some Elp1 heterozygous embryos were used as control. The mouse SCGs at E14.5 were isolated and cultured as previously described[14,93]. In brief, dissected SCGs were washed once by 1× HBSS and centrifuged at $300 \times g$ for 5 min. SCGs were incubated with 0.25 mg/ml of trypsin (Thermo Fisher Scientific, 25200056) at 37 °C for 15 min. After centrifugation, 1 mg/ml of trypsin inhibitor (Thermo Fisher Scientific, R007100) was added to SCGs and they were incubated for another 5 min at 37 °C. SCGs were then centrifuged and fully dissociated by pipetting with SCG culture medium that consists of Neurobasal medium, B27, GlutaMAX (according to the manufacturer's instructions) and 60 ng/ml NGF (5–10 μl/2 SCGs). Dissociated SCGs were plated on 96-well Bio-Circuit or CytoView MEA plate (Axion BioSystems, M768-BIO-96 or M768-tMEA-96W), that were coated with PO/LM/FN, as droplets (5–10 μl/2 SCGs each well) for MEA measurements or on regular cell culture plates (Corning, 3524) for RNA sampling or immunostaining. For MEA plates, droplets were incubated at 37 °C for 30–45 min, and filled up with 200 μl culture medium per well. 10 μM Y-27632 can be added to culture medium on the day of plating for improved attachment. Animal protocol IACUC #2021-35–81 (Lefcort, Montana State University), #A2019 05-013 (Liu, University of Georgia).

## Live cell flow cytometry

Day 10 NCCs were washed with 1× PBS and dissociated in accutase. The cell/accutase mixture was washed by mixing the solution with FACS buffer that contains 1× DMEM (Life Technologies, 10829-018), 2% FBS (Atlanta Biologicals, S11150) and 200 mM L-Glutamine. The cells were spun twice at $200 \times g$ for 4 min. Count the cells and incubate $1 \times 10^6$ cells with mouse anti-CD49d-PECy7 (Biolegend, 304313, 1:20 in 100 μl) for 20 min on ice. After incubation, wash cells by mixing the mixture with FACS buffer and spinning twice at $200 \times g$ for 4 min. Cells in FACS buffer were kept on ice and analyzed using Beckman Coulter CytoFLEX.

## Quantitative RT-qPCR

Cell lysates were obtained from 1 to 2 wells of a 24-well plate for each sample and the RNA was extracted using the Trizol reagent (Invitrogen, 15596026). 1 μg of total RNA each sample was reverse-transcribed into cDNA using iScript™ Reverse Transcription Supermix (Bio-Rad, 170884) according to the manufacturer's instruction. RT-qPCR was performed with SYBR green (Bio-Rad) and analyzed with a CFX96 Touch Deep Well Real-Time PCR Detection System (Bio-Rad). The primers used are listed in Supplementary Table 1.

## Immunofluorescence staining

Neurons cultured in the desired culture plates were fixed with 4% paraformaldehyde for 20 min and washed with 1× PBS. Samples were permeabilized using 0.3% Triton, 1% BSA and 3% goat or donkey serum in 1x PBS for 20 min. Primary antibodies were added to the cells and incubated at 4 °C overnight. The next day, cells were washed twice and incubated with secondary antibodies (at 1:400 dilution) for 1 hr at RT. Images were taken using the Lionheart FX Automated Microscope. The antibodies used here are listed in Supplementary Table 2. Secondary antibodies used in this study are goat anti mouse AF488 (H+L)(Invitrogen, A11029), goat anti mouse AF647 (H + L)(Invitrogen, A21235), goat anti mIgG1 AF488 (Invitrogen, A21121), goat anti mIgG1 AF647 (Invitrogen, A21240), goat anti rabbit (H + L) AF647 (Invitrogen, A21245), goat anti rabbit (H + L) AF488 (Invitrogen, A11008), goat anti chicken AF488 (Invitrogen, A-11039).

## Western blot analysis

For protein lysate collection, day 35 symNs in one well of 6-well culture plate (Corning) were prepared. After washing with 1× PBS, cells were lysed by lysis solution that contains 1× RIPA buffer (Sigma-Aldrich, R0278), 1 mM PMSF protease inhibitor (Thermo Fisher Scientific, 36978) and 10× PhosSTOP phosphatase inhibitor (Sigma-Aldrich, 4906845001). Bradford reagent (Bio-Rad, 5000006) was used for measuring protein concentration. For western blotting, 20 μg/well of total protein for each sample was loaded and ran in 12% acrylamide gels. Proteins were then transferred to nitrocellulose membrane and blocked with 5% skim milk in TBST. The membranes were incubated with primary antibodies including mouse monoclonal anti-hNET (Mab Technologies, NET17-1, 1:1000) and mouse monoclonal anti-c-Fos (Abcam, ab208942, 1:1000) at 4 °C overnight. Next day, the membranes were washed 3 times with PBST and incubated with goat anti-mouse and goat anti- HRP for 1 h at RT.

## MEA assay

SymNs for the extracellular spike measurements were plated on 96-well BioCircuit or CytoView MEA plate (Axion BioSystems), coated with PO/LM/FN, at $1 \times 10^5$ cells/cm² density. If BioCircuit plates were used, it is recommended that additional wells on a regular cell culture plate with visible bottoms (Corning, CLS35) are used at the same condition to monitor the growth of symNs. Neural activity was measured at the desired timepoints using the Maestro Pro multiwell plate reader under neural detection mode according to manufacturer's instructions.

## Calcium imaging

SymNs for the experiment were plated on 96-well black well/clear bottom plates (Corning, 3603), coated with PO/LM/FN, at $1 \times 10^5$ cells/cm² density. Day 35 symNs were washed with 1× HBSS (Gibco) once. For the blank, 200 μl of fresh culture medium was added into one well of each experimental group. For intracellular $Ca^{2+}$ tracing, 2 μM of Fluo-4 AM (TOCRIS, 6255) was made with culture medium and added to duplicated experimental groups (200 μl/well). SymNs with Fluo-4 were incubated at 37 °C for 20 min, followed by a 1× HBSS wash three times and incubated for 30 min with fresh medium. After the incubation, symNs were read using a fluorescence microplate reader (BioTek)

immediately at 440 nm excitation and 520 nm emission or imaged using the Lionheart FX Automated Microscope.

## Norepinephrine imaging

The NE tracer (NS510) was a kind gift by Timothy Glass's laboratory at University of Missouri, and used according to previous publication[63]. SymNs for the experiment were plated on 96-well black well/clear bottom plates (Corning), coated with PO/LM/FN, at $1 \times 10^5$ cells/cm$^2$ density. Day 35 symNs were washed with 1× HBSS once. For the blank, 200 μl of fresh culture medium was added into one well of each experimental group. For intracellular NE tracing, 0.5 μM of NS510 was made with culture medium and added to duplicated experimental groups (200 μl/well). SymNs with NS510 were incubated at 37 °C for 45 min, followed by a 1× HBSS wash twice and read using a fluorescence microplate reader (BioTek) immediately at 440 nm excitation and 520 nm emission or imaged using a Lionheart FX Automated Microscope.

## NET reuptake assay

The assay was performed according to manufacturer's instructions (Neurotransmitter Transporter Uptake Assay Kit, Molecular Devices, R6138). SymNs for the experiment were plated on 96-well black well/clear bottom plates (Corning), coated with PO/LM/FN, at $1 \times 10^5$ cells/cm$^2$ density. Day 35 symNs were washed with 1× HBSS once. For the blank, 200 μl/well of 1×HBSS was added in to one well of each experimental group. For NET reuptake measurement, 100 μl of 1× HBSS with 100 μl dye solution/well was added in to 2 wells (duplication is recommended) of each experimental group. The plate was read immediately using a fluorescence microplate reader (BioTek) at 440 nm excitation and 520 nm emission.

## NE and ACh ELISA

The assays were performed according to manufacturer's instructions (NE: EagleBio, NOU39-K01; ACh: Sigma, MAK056). Media in two wells of a 24-well of day 35 symNs from each experimental group was changed (500 μl/well) one day before the experiment. 24 h later, media from two wells were harvested and pooled together to concentrate the NE. To avoid NE degradation, sample stabilizer included in the kit was added to each sample. Collected media was spun at $300 \times g$ for 5 min to remove debris. The samples were ready for NE detection or were stored at −80 °C for long-term storage.

## MTT assay

PrestoBlue Viability Reagent (Invitrogen, A13261) was used for measuring symN viability. SymNs for the experiment were plated on 96-well cell cultured plates, coated with PO/LM/FN, at $1 \times 10^5$ cells/cm$^2$ density (two wells for each experimental group). Media on day 35 neurons was replaced with 1:10 diluted viability reagent with fresh culture medium (200 μl per well, duplication is recommended), and incubate at 37 °C for 3 h. 200 μl of culture medium without dye solution was added to one well of each group for blank measurement. After 3 h, 100 μl of medium and dye mixture of each well was transferred to 96-well plates with clear flat-bottom (Corning, 3596), and read using a microplate reader (BioTek) immediately at 570 and 600 nm absorbance.

## RNAseq and GO term analysis

RNA was purified as previously described. RNA concentration and integrity was assessed by Bioanalyzer. Libraries were prepared using KAPA's Stranded mRNA-seq kit (#KK4821) with halved reaction volumes. During library preparation, mRNA was selected using oligodT beads, followed by RNA fragmentation and cDNA generation using random hexamer priming. The number of cycles used in the library PCR was determined based on kit recommendations. Qubit and fragment analyzer were used to determine library concentration and the size distribution of the library, respectively. RNA libraries were prepared for sequencing using Illumina's Dilute and Denature protocol. Libraries were then pooled and diluted to 2 nM followed by denaturation using NaOH. The denatured library was further diluted to 2.2pM, and PhiX was added at 1% of the library volume. RNA pools were run on an Illumina NextSeq 2000. Sequenced reads were demultiplexed and adapter and barcode sequences were trimmed in BaseSpace. All analyses were done on the Galaxy web platform (*usegalaxy.org*)[94]. Quality control of the reads were assessed using FastQC (version 0.11.9) (http://www.bioinformatics.babraham.ac.uk/projects/fastqc/). Low quality bases were trimmed from sequencing reads using Trimmomatic (version 0.38)[95]. Reads were then mapped to the human genome (GRCh38.p13) using HISAT2 (version 2.21)[96]. Quality control of the aligned reads was performed using QualiMap BamQC (version 2.2.2c)[97,98]. Reads per gene were counted using HTseq (version 0.91)[99] with a minimum alignment quality value of 10. The raw count matrix was then processed using DESeq2 (version 1.34.0)[100] with default settings to analyze differential expression between sample groups, perform principal component analysis, and measure sample-to-sample distances. Genes that were significantly downregulated (p-adj <0.05) were subjected to gene ontology analysis using the DAVID functional annotation tool (https://david.ncifcrf.gov/). Significant (FDR < 0.05) GO terms were plotted.

## Statistical analysis

Data was collected from at least (or more) three independent experiments (biological replicates), with multiple technical replicates each. Biological replicates are defined as[41] independent experiments conducted several days apart either started from a new frozen vial of that particular cell line[101] or started from a consecutive passage number of that cell line. Multiple clones derived from one patient line have been analyzed previously[29] and there were shown to not have significant variability, thus here we used one clone per iPSC line. Quantification results were presented as mean ± SEM. Significance calculation between two experimental groups was analyzed by two-tailed Student's t-test. For multiple group comparison, one-way ANOVA is used followed by Tukey's multiple comparison tests, while two-way ANOVA followed by Šídák multiple comparisons was used for comparisons among different timepoints. All analysis is performed using Prism 9.

## Statistics and reproducibility

Data of Fig. 1f, i can be reproduced in every biological replicate presented in the paper. Figure 3e, f, h have been confirmed in two litters of mouse embryos. Figure 4f has been confirmed in two litters of mouse embryos (one WT and one KO per litter). Figure 4e, n have been repeated in three biological replicates. Supplementary Fig. 1d–d', h have been repeated in three biological replicates. Supplementary Fig. 2b, c have been repeated in three biological replicates. In Supplementary Fig. 2f, three biological replicates of symNs were compared to one biological replicate of SN as the positive control, and a picture from one of the symNs was representatively selected. Supplementary Figs. 3a, 4b, 5a, d have been confirmed in three biological replicates. Figure 6c has been confirmed in two litters of mouse embryos (one WT and one KO per litter). Supplementary Fig. 7b has been confirmed in three biological replicates.

## Reporting summary

Further information on research design is available in the Nature Portfolio Reporting Summary linked to this article.

# Data availability

All data generated or analyzed in this study are included in this published article and its supplementary data file. Raw data points and uncut western blot gel data generated in this study have been deposited in the figshare.com database under accession code: https://

doi.org/10.6084/m9.figshare.21394737 (https://figshare.com/authors/Nadja_Zeltner/14007450). FASTQ files for RNA sequencing data was deposited through NCBI Gene Expression Omnibus, accession number GSE212255. Source data are provided with this paper.

## Code availability

Data collection was done using the following softwares: LAS X, Gen5.05, Axion AXIS Navigator, Axion NeuralMetric Tool, iBright Imaging System, CFX96 Real-time System, CytExpert 2.0. Data analysis was done using the following software: Prism 9, Fiji (ImageJ), CFX Maestro, FlowJo10.5.3.

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

## Acknowledgements

We thank Dr. Timothy Glass and his group for providing us with the NS510 probe, and Dr. Fikri Avci and his group for access to equipment for fluorescence ELISA measurements. We also thank Dr. James Lauderdale, Dr. Stephen Dalton, Dr. Jesse Shank, Dr. Chia Wei Huang and Zhonghou Wang for critical discussions and technical support. This work was funded by faculty start-up funds from the University of Georgia to N.Z. and NIH/NINDS 1R01NS114567-01A1 to N.Z., NIH/NIDCD R21DC018089 to HX.L., and RO1 DK117473 to F.L. Parts of the cartoons in the figures (Figs. 2i, 3d, 4a, 6 and Supplementary Figs. 1a, 6a, 8a) were created with motifolio.com and/or BioRender.com.

## Author contributions

H.F.W. conceived, conducted and analyzed experiments, wrote on manuscript. W.Y. and J.C. conducted animal tissue processing. K.S.D. analyzed RNA seq. data and provided advice. F.L. and H.X.L. provided support, advice and mice for animal experiments. C.W.H. conducted western blot experiments. G.H. provided funds and oversight for C.W.H. N.Z. conceived, designed, and lead the study, provided financial and administrative support, edited and approved the manuscript.

## Competing interests

The authors declare no competing interests.
