## [Peer Review File · Nature Communications]

Norepinephrine transporter defects lead to sympathetic hyperactivity in Familial Dysautonomia modelsREVIEWER COMMENTS

Reviewer #1 (Remarks to the Author):

The team has developed a new way to generate sympathetic neuron from hPSCs, then applied to FD modeling. They found that FD sympathetic neurons show intrinsically hyperactive in vitro culture, and also similar features when co-cultured with CM and in vivo FD model. Further they found that the hyperactivities are still present in the neurons derived from isogenic PSC control, which is puzzling. In addition, they found decreased levels of intracellular NE, NE uptake, excessive extracellular NE in FD neurons, suggesting the reason of hyperactivity. Finally they found several drugs can reverse such neuronal hyperactivity.

I think it is an interesting study, and it has three novel points, including modeling an autonomic disease with a new sympathetic neuron protocol, uncovering a previously unexplored disease mechanism of FD, and identifying potential drug candidates for the patients. However, several points might need to be revisited.

Major points

It has not been studied well how human PSCs can develop into sympathetic and parasympathetic neurons. This issue is important to model autonomic neuron diseases like FD, because the patients present complex symptoms related with autonomic nervous system. Did authors have any data to support their final neurons are solely sympathetic neuron lineage? Is there any parasympathetic neuron inclusion?

What could be an in vivo counterpart of the sympathoblasts? How we can interpret the Fig. 1h data? Does that mean the FD sympathoblasts are dying during development or postnatal stage?

Did any drug in Fig. 5 rescue the IKBKAP splicing aberration? Otherwise what could be the mechanism of the rescuing hyperactivities?

Minor points

Interesting data in Fig. 2J. It is very challenging to trace which neurons are connected to each CM without antibody or dye staining. In fact, when the neurons and CMs are co-cultured, there are many CMs in multiple spots connected to many different neurons in a well. Did the authors use a special tissue culture plate? I can only find 'For symN and CM co-culture, day 14 symN progenitors were added on top of day 15 CMs at 1×10^5 cells/cm² density in symN medium plus CM medium at a 1:1 ratio.' In M&M section. It is difficult for me to find how the authors confirmed the physical connection between the symN and CM, and how many CM colonies they counted in one repeat. It will be great to include this information for other researchers.

In Fig. 1d, it is difficult to see the cell morphology in the small figure. Please move the "PHOX2B" out of the image.

What is 'explain in figure legend' in purple color in Fig. 4g?

Reviewer #2 (Remarks to the Author):

In this elegant study Zeltner's team show that sympathetic neurons derived from human pluripotent stem cells of patients with Familial Dysautonomia are spontaneously hyperactive and have reduced activity of the norepinephrine transporter. Multi-electrode-array recordings show increased spike frequency compared to sympathetic cells derived from controls. Interestingly, hyperactivity only occurred in peripheral sympathetic neurons, not in cortical neurons derived from FD iPSC lines. The hyperactivity was confirmed when FD sympathetic neurons were cocultured with cardiomyocytes, resulting in a higher beating rate than when cocultured with sympathetic neurons from controls. Hyperactivity was also seen in sympathetic neurons of the superior cervical ganglia of FD mouse model at the embryonic stage.

The authors performed several experiments to understand the mechanism of neuronal hyperactivity. Tetrodotoxin did not alter hyperactivity and there was no difference in expression of nicotinic or Gaba receptors, or potassium or calcium channels. Importantly, NET transporter expression was reduced, leading to increase NE concentration in extracellular space. A similar finding occurred in control neurons treated with the NET inhibitor nomifensine.

Finally, the authors tested dexmedetomidine and carbidopa, two drugs used clinically to treat sympathetic hyperactivity in patients with FD, and confirm their ability to reduce excessive firing in their cell model.

These findings are crucial to understand the phenotype and should help drug development for symptomatic therapy. Until now, the explanation for the excessive sympathetic activity in FD patients was lack of afferent baroreflex restrain. Intrinsic neuronal sympathetic hyperactivity and reduced NET expression add a robust second mechanism that further explains the phenotype.

Assessments of sympathetic neuronal function in humans is possible with microneurography. Of interest to the authors, microneurography was performed in FD patients (Macefield J of Physio 2011). It showed spontaneous desynchronized sympathetic nerve activity, the neurons do not fire with the characteristic bursting pattern in between heart beats, rather tending to fire continuously during periods of emotional arousal. In other words, sympathetic neurons that control blood pressure are intact but are deprived of the normal baroreceptor inputs that constrain them to fire in the intervals between heart beats.

The findings of Zeltner et al further explain the patients heightened sympathetic activity, and paroxysmal episodes of tachycardia, hypertension, retching and vomiting that can last from hours to weeks. Medical illnesses or emotions can trigger this "autonomic crises" so that patients are literally at the mercy of their emotions

A defect in the NET transporter also explains the puzzling finding of lack of fluorodopamine uptake by sympathetic post ganglionic neurons innervating the myocardium in patients with FD, which was not in line with clinical findings. It had been interpreted as sympathetic denervation but was likely the result of impaired NET transporter activity.

In sum, this work is a crucial addition to understanding the FD phenotype.

Horacio Kaufmann

Reviewer #3 (Remarks to the Author):

In this study the authors used patient-derived human pluripotent stem cells (hPSC) to characterize the sympathetic neuron (symN)-specific disease mechanism underlying FD autonomic dysfunction. They demonstrated that FD symNs are intrinsically hyperactive and that a defective norepinephrine autoregulatory pathway underlies the observed hyperactivity. They then used their well-characterized cellular model system to screen drugs based on their ability to lower symN hyperactivity. While this manuscript is a good example of how patient-derived iPSC cellular models can be used to gain insights into disease molecular mechanisms and to identify novel potential therapies for patients, the fact that the observed hyperactive phenotype is not rescued by genetic correction of the most common ELP1 splicing mutation raises critical questions about the validity of these findings for the broader FD population.

Major comments:

1) In the introduction, the authors state: "In fact, it's been shown that FD patients with the most severe symptoms harbor a modifier mutation in the gene LAMB4, encoding laminin β 4". This sentence is simply not true and it overstates the findings in Zeltner et al 2016. In this study only a couple FD patient cell lines were analyzed for mutations in the LAMB4 gene. There has been no rigorous genotype-phenotype study performed in patients that validates the link between the LAMB4 mutations and disease severity. This statement must be restated to better reflect the findings in Zeltner et al 2016.

2) The authors should check if the numbers of the biological replicates in the figure legends are correct. There is inconsistency between the numbers reported in the Results section compared with the n in the figure legends. In the Results section the authors stated using two different FD patient cell lines (iPSC-FD-S2 and iPSC-FD-S2) and two control lines (iPSC-ctrl-C1 and hESC-ctrl-H9), but in the figure legends they reported many more biological replicates (n=4-9). Please provide an explanation.

3) The fact that the correction of the FD major splicing mutation in the iPSC-FD-S2 line using CRISPR-cas9 does not result in the rescue of the hyperactive phenotype, raises critical questions about the validity of this cellular model in accurately recapitulating the disease pathogenesis. The authors reference the role of ELP1 in translation, but ELP1 also has a well-established role in transcriptional elongation. Is the hyperactive phenotype identified in these cell types potentially independent from or secondary to reduction of ELP1? If this phenotype is the result of a mutation in a modifier gene, the authors should first demonstrate that the mutation of the modifier gene is associated with disease severity, and then establish the link between this mutation and the hyperactive phenotype. This study is based on two FD lines that both happen to have a mutation in the LAMB4 gene. How many FD patients have a mutation in the LAMB4 gene? What is the frequency of this mutation in the general population and in the FD population? Did the authors correct the ELP1 mutation also in the iPSC-FD-S3 cell line to confirm no phenotypic rescue? This is a critical finding that certainly needs further investigation in order to demonstrate that this specific phenotype is relevant for FD pathogenesis and not coincidental.

4) In the discussion, the authors speculate that a potential explanation for the no observed rescue of the symN hyperactivity after correction of the FD major mutation could be attributed to "the downstream consequence of altered developmentally-timed genetic programs due to the loss of ELP1 expression and hence they cannot be re-set once the nervous system is mature." The authors should clarify this point. Isn't the mutation corrected at the iPSC stage? If yes, why are they referring to mature nervous system here? If not, they should clarify at which stage of the neuronal differentiation protocol the ELP1 mutation was corrected.

Minor comments:

1) In the Abstract (lines 21-22), the authors state: "Familial dysautonomia (FD) is a rare neurodevelopmental and neurodegenerative disorder that affects the sympathetic nervous system."

More precisely FD affects both the sensory and sympathetic nervous system.

2) In the Introduction (line 42), the word “complicated” should be replaced with “complex” since this adjective is more appropriate in this context.

3) In the introduction (lines 77-78 and lines 152-153) there are citations missing that describe the characterization of sympathetic ganglia volume in mouse models of FD, several studies have been published.

4) The authors should use correct capitalization to distinguish human from mouse, ELP1, Elp1

5) In line 266 one bracket is missing.

Point-by-point rebuttal for Wu et al., Nature Communications:

First, we would like to thank the editor and each of the reviewers for the valuable comments on our manuscript, for their recognition that this work is important for the field, and for giving us the opportunity to prepare a revised version of our work that integrates all the feedback.

We agree with the main assessment of the reviewers and editors that a major revision will strengthen our work. Thus, in this rebuttal we will first (section 1) outline the major additions we have made to the manuscript during this revision, followed by a point-by-point rebuttal (section 2) to each of the reviewers' comments.

Changes in the manuscript and novel data are highlighted in blue in the revised manuscript and the below point-by-point rebuttal for easy reference. We also list the page and line numbers, where specific changes can be found in the revised manuscript.

Section 1: Major additions to the story

- We conducted a deeper genetic markers analysis to distinguish symNs from parasymNs and show the purity and lack of parasymN contamination in our symN cultures (Rev 1, **Sup. Fig. 2f, g**)
- We addressed the mechanism of action of some tested drugs (Rev 1, **Fig. 5j,k**)
- We incorporated clinical relevance into the introduction and discussion (Rev 2)
- We provide a deeper assessment of the role of *ELP1* in symN hyperactivity (Rev 1, 3, **Sup. Fig. 7**)
- We assess the potential role of *LAMB4* modifier mutations in symN hyperactivity (Rev 3, **Sup. Fig. 7**)
- We added several FD and control iPSC lines and assessed their symN hyperactivity phenotype (Rev 1, 3, **Sup. Fig. 7**)
- We reprogrammed a new FD iPSC line (iPSC-FD-M4), added a heterozygous carrier FD line (iPSC-carrier-A1) and obtained an additional healthy control iPSC line (iPSC-ctrl-652), the donor of which matches the original FD lines in age closer. We assessed these lines' symN hyperactivity phenotype (Rev 3, **Sup. Fig. 7**)
- We reproduced the defect in NGF retrograde transport described in the FD mouse¹, in our model to strengthen the validity of our cellular model in reproducing phenotypes (Rev 3, **Sup. Fig. 8**)
- We found that the link between the FD *ELP1* mutation, NET deficiencies and symN hyperactivity may be Rab proteins (Rev. 1, 3, **Fig. 4m,n** and **Sup. Fig. 9**)
- We conducted a bulk RNA seq. experiment to assess differential expression of FD and control symNs and confirmed the important role of Rab proteins (**Sup. Fig. 9**)

Section 2: Point-by-point response to reviewer's suggestions

Reviewer #1. The team has developed a new way to generate sympathetic neurons from hPSCs, then applied this to FD modeling. They found that FD sympathetic neurons show intrinsically hyperactive in vitro culture, and also similar features when co-cultured with CM and in vivo FD model. Further they found that the hyperactivities are still present in the neurons derived from isogenic PSC control, which is puzzling (please see our responses to reviewer #3 for this point). In addition, they found decreased levels of intracellular NE, NE uptake, excessive extracellular NE in FD neurons, suggesting the reason of hyperactivity. Finally, they found several drugs can reverse such neuronal hyperactivity.

I think it is an interesting study, and it has three novel points, including modeling an autonomic disease with a new sympathetic neuron protocol, uncovering a previously unexplored disease

mechanism of FD, and identifying potential drug candidates for the patients. However, several points might need to be revisited.

Major points:

- It has not been studied well how human PSCs can develop into sympathetic and parasympathetic neurons. This issue is important to model autonomic neuron diseases like FD, because the patients present complex symptoms related with autonomic nervous system. Did authors have any data to support their final neurons are solely sympathetic neuron lineage? Is there any parasympathetic neuron inclusion? We would like to thank this reviewer for raising this important question. From the clinical standpoint, involvement of the parasympathetic nervous system in the FD pathology is rather unclear. Particularly, because there are contradicting and opposing reports from the clinic²⁻⁵. Because of that, and because of the power of our hPSC system, our lab is very interested in assessing parasympathetic involvement in FD. We have early projects ongoing in the lab to generate parasympathetic neurons and assess FD phenotypes. However, we believe that this data is beyond the scope of the current work and would not be ready within the revision time. Nevertheless, we wanted to address the reviewers concern and show more characterization of our sympathetic neuron differentiation protocol to distinguish the neurons from parasympathetic neurons. **Sup. Fig 2a** (red bars) now shows that 75% of cells in our cultures are neurons and of those neurons > 90% express the symN marker TH, thus the cultures are highly enriched. **Sup. Fig 2f** shows that the parasympN marker ChAT is essentially absent and **Sup. Fig. 2g** shows that the neurons release ~7 times more NE than ACh. Of further note is that there are some symNs that release ACh *in vivo*⁶.

Please see revised manuscript page 7, line 147

- What could be an *in vivo* counterpart of the sympathoblasts? How can we interpret the Fig. 1h data? Does that mean the FD sympathoblasts are dying during development or postnatal stage? Previous studies using an FD mouse model identified decreased neural survival in E12.5 sympathetic ganglia as well as reduced sympathetic ganglion size, while NCC survival was intact^{7,8}. We believe that our data recapitulates these findings relatively well. It is rather difficult to correlate *in vivo* and *in vitro* timing on top of mouse versus human timing. However, we would estimate that what we call the human sympathoblast in our hPSC system might correlate with immature sympathetic neurons around E12.5 or slightly before in the mouse. In our hPSC system, we do not reach an equivalent of the postnatal stage, since that would be many months of *in vitro* culture, thus all our data represents the developmental stage. In patients however, it is known that symNs degenerate/die in the postnatal and adult stage, evident by autopsy findings⁹ and the fact that symptoms get worse over time, such as dysautonomic crisis' and anxiety become more frequent¹⁰.

Please see revised manuscript page 9, line 189

- Did any drug in Fig. 5 rescue the *IKBKAP* splicing aberration? Otherwise, what could be the mechanism of the rescuing hyperactivities? We thank the reviewer for this important question. We have now assessed two of the drugs from Fig. 5 that rescued FD symN hyperactivity, Dexmed and CNO, and assessed if they affect *ELP1* splicing as well as NET expression in FD. We found that neither of them rescues *ELP1* splicing levels (**Fig. 5j**), which is consistent with our previous data showing that the genetic rescue of the *ELP1* mutation does not reverse the hyperactivity (**Sup. Fig. 7a**). We thus further looked into the effects of CNO on NET levels (**Fig. 5k**) and found that treatment with CNO rescues NET expression levels after 48 or 72 hours, suggesting that its mechanism of

action to rescue symN hyperactivity is via elevation of NET expression levels.
Please see revised manuscript page 19, line 431

Minor points:

- Interesting data in Fig. 2J. It is very challenging to trace which neurons are connected to each CM without antibody or dye staining. In fact, when the neurons and CMs are co-cultured, there are many CMs in multiple spots connected to many different neurons in a well. Did the authors use a special tissue culture plate? I can only find 'For symN and CM co-culture, day 14 symN progenitors were added on top of day 15 CMs at 1×10^5 cells/cm² density in symN medium plus CM medium at a 1:1 ratio.' In M&M section. It is difficult for me to find how the authors confirmed the physical connection between the symN and CM, and how many CM colonies they counted in one repeat. It will be great to include this information for other researchers. We thank the reviewer to highlight this point. We always strive to detail our experimental set ups in a way to increase reproducibility by other researchers. Thus, we now have added more details about the symN-CM co-cultures in the material section. We also added IF staining, with a-ACTININ and PRPH and SYP to show the neuron-CM junction better (Sup. Fig. 5d).
Please see revised manuscript page 38 (in Materials&Methods) and page 11, line 233 in the manuscript
- In Fig. 1d, it is difficult to see the cell morphology in the small figure. Please move the "PHOX2B" out of the image. We assume that the reviewer meant Supp. Fig. 1d here. We have now moved the inset outside and made it a separately standing, bigger image for easier view.
Please see, Sup. Fig. 1d, right.
- What is 'explain in figure legend' in purple color in Fig. 4g? We apologize for this oversight. We have removed this writing now.
Please see Fig. 4g.

Reviewer #2. In this elegant study Zeltner's team show that sympathetic neurons derived from human pluripotent stem cells of patients with Familial Dysautonomia are spontaneously hyperactive and have reduced activity of the norepinephrine transporter. Multi-electrode-array recordings show increased spike frequency compared to sympathetic cells derived from controls. Interestingly, hyperactivity only occurred in peripheral sympathetic neurons, not in cortical neurons derived from FD iPSC lines. The hyperactivity was confirmed when FD sympathetic neurons were cocultured with cardiomyocytes, resulting in a higher beating rate than when cocultured with sympathetic neurons from controls. Hyperactivity was also seen in sympathetic neurons of the superior cervical ganglia of FD mouse model at the embryonic stage.

The authors performed several experiments to understand the mechanism of neuronal hyperactivity. Tetrodotoxin did not alter hyperactivity and there was no difference in expression of nicotinic or Gaba receptors, or potassium or calcium channels. Importantly, NET transporter expression was reduced, leading to increase NE concentration in extracellular space. A similar finding occurred in control neurons treated with the NET inhibitor nomifensine.

Finally, the authors tested dexmedetomidine and carbidopa, two drugs used clinically to treat sympathetic hyperactivity in patients with FD, and confirm their ability to reduce excessive firing in their cell model.

- These findings are crucial to understand the phenotype and should help drug development for symptomatic therapy. Until now, the explanation for the excessive sympathetic activity in FD patients was lack of afferent baroreflex restraint. Intrinsic neuronal sympathetic hyperactivity and reduced NET expression add a robust second mechanism that further explains the phenotype. We would like to thank reviewer 2 and share their enthusiasm. We now have added similar wording to the discussion.
Please see manuscript Page 25, line 566
- Assessments of sympathetic neuronal function in humans is possible with microneurography. Of interest to the authors, microneurography was performed in FD patients (Macefield J of Physio 2011). It showed spontaneous desynchronized sympathetic nerve activity, the neurons do not fire with the characteristic bursting pattern in between heart beats, rather tending to fire continuously during periods of emotional arousal. In other words, sympathetic neurons that control blood pressure are intact but are deprived of the normal baroreceptor inputs that constrain them to fire in the intervals between heart beats.
The findings of Wu et al. further explain the patients heightened sympathetic activity, and paroxysmal episodes of tachycardia, hypertension, retching and vomiting that can last from hours to weeks. Medical illnesses or emotions can trigger this “autonomic crises” so that patients are literally at the mercy of their emotions. We have now used this information to enhance our introduction.
Please see manuscript Page 4, line 74
- A defect in the NET transporter also explains the puzzling finding of lack of fluorodopamine uptake by sympathetic post ganglionic neurons innervating the myocardium in patients with FD, which was not in line with clinical findings. It had been interpreted as sympathetic denervation but was likely the result of impaired NET transporter activity.
In sum, this work is a crucial addition to understanding the FD phenotype. We are very excited to hear the reassuring comments made by reviewer 2 and want to thank their insightful analysis of our work and important comparison to the clinic. We have made several changes throughout the manuscript to employ and incorporate reviewer 2’s wording and knowledge into our work. We much appreciate these additional comments and believe that they made our work stronger, more relevant and more connected to clinical work.
Please see revised manuscript page 26, line 581

Reviewer #3. In this study the authors used patient-derived human pluripotent stem cells (hPSC) to characterize the sympathetic neuron (symN)-specific disease mechanism underlying FD autonomic dysfunction. They demonstrated that FD symNs are intrinsically hyperactive and that a defective norepinephrine autoregulatory pathway underlies the observed hyperactivity. They then used their well-characterized cellular model system to screen drugs based on their ability to lower symN hyperactivity. While this manuscript is a good example of how patient-derived iPSC cellular models can be used to gain insights into disease molecular mechanisms and to identify novel potential therapies for patients, the fact that the observed hyperactive phenotype is not rescued by genetic correction of the most common ELP1 splicing mutation raises critical questions about the validity of these findings for the broader FD population.

Major comments:

- In the introduction, the authors state: “In fact, it’s been shown that FD patients with the most severe symptoms harbor a modifier mutation in the gene *LAMB4*, encoding laminin β 4”. This sentence is simply not true and it overstates the findings in Zeltner et al 2016. In this study only a couple FD patient cell lines were analyzed for mutations in the *LAMB4* gene. There has been no rigorous genotype-phenotype study performed in patients that validates the link between the *LAMB4* mutations and disease severity. This statement must be restated to better reflect the findings in Zeltner et al 2016. We thank reviewer 3 for addressing this issue. While we disagree with their statement ‘This sentence is simply not true’, we do agree that our wording was not careful enough for the current state of the literature. Thus, we have now moved and re-written this section in the introduction to clarify previous findings about *LAMB4* modifier mutations found in some FD patients (page 5, line 100). Reviewer 3 is correct, in that there has not been a rigorous genotype-phenotype study published yet to assess modifier mutations in the larger FD population and we do agree that this is a much-needed analysis in the field. We can say that our lab and collaborators in the field are currently working on such a study. However, these results are not ready for publication for a while. More importantly, for the current study of hyperactivity in FD sympathetic neurons, we do believe that they would not be a strong fit here and would distract the focus of the underlying study. Lastly, we believe that this is outside the scope of the current study. However, we now addressed the concern of the potential *LAMB4* modifier mutation influence on the symN hyperactivity phenotype with new experimental work. We used three additional FD-iPSC lines derived from FD patients with mild symptoms and a mild FD phenotype in the cellular model¹¹. One of them iPSC-FD-M4 was reprogrammed newly during this revision period. We found that these three lines’ symNs were also hyperactive (**Sup. Fig 7d**). Mild lines have the genotype *ELP1*^{-/-}, *LAMB4*^{+/+}, thus this suggests that we can rule out that symN hyperactivity is a result of the *LAMB4* mutation. We also added a new healthy control iPSC line (iPSC-ctrl-652), the donor of which better matches the age of our FD patients and found it to have physiological electric activity (**Sup. Fig 7c**). We believe that these additions strengthen the results from our cellular model significantly.
Please see revised manuscript page 13, line 282

- The authors should check if the numbers of the biological replicates in the figure legends are correct. There is inconsistency between the numbers reported in the Results section compared with the n in the figure legends. In the Results section the authors stated using two different FD patient cell lines (iPSC-FD-S2 and iPSC-FD-S3) and two control lines (iPSC-ctrl-C1 and hESC-ctrl-H9), but in the figure legends they reported many more biological replicates (n=4-9). Please provide an explanation. We want to thank reviewer 3 for bringing up this point. Standard in the field of hPSCs is that biological Ns are defined as independently conducted differentiations. This can be done from a freshly thawed vial of cells or from a consecutive passage number; either way the differentiation has to be conducted several days apart to be considered a biological replicate (N)¹². Sometimes researchers use multiple clones reprogrammed from one patient to verify that there is little variability among clones from one patient. We have done that for the patient lines used here in the Zeltner 2016 study¹¹, where the lines were established and used first. There, we discovered that the variability between clones derived from the same patient is minimal and thus have combined the data of the clones for each patient or continued out work with one of the clones. Due to that validation in 2016, here we used one clone per patient line. However, we would like to clarify here that biological Ns, by definition are not patient lines. It is not feasible to conduct experiments with that many patient lines on top of at least 3 independent differentiation repeats. Thus, when indicated in the figure legends 3-5 N’s, this represents 3 to 5 differentiations conducted

several days apart with this clone of this patient. We have now added this explanation in a special section in the materials and methods for clarification. Furthermore, we checked every figure to assure that the biological Ns (as defined this way) are indicated correctly. Lastly, we have now added 1 more FD (iPSC-FD-M4) patient iPSC line and 1 more control iPSC (iPSC-ctrl-652) line to our results for added rigor, please see more about these results in the **Sup. Fig. 7** and the next point below.

Please see revised manuscript page 46, line 1024 (Materials&Methods) and page 13, line 283

- The fact that the correction of the FD major splicing mutation in the iPSC-FD-S2 line using CRISPR-cas9 does not result in the rescue of the hyperactive phenotype, raises critical questions about the validity of this cellular model in accurately recapitulating the disease pathogenesis. We agree with the reviewer that the result that the iPSC-*ELP1*_{rescue}-T6 line does not rescue the symN hyperactivity is puzzling. We have now conducted additional work to address this point further. In **Sup. Fig. 7**, we first added several new FD and control iPSC lines to strengthen our results. The new healthy control (iPSC-ctrl-652) symNs were not hyperactive. We then employed an iPSC line from a FD carrier (iPSC-carrier-A1), who is heterozygous in *ELP1* and in *LAMB4*, thus it has the same genotype as iPSC-FD_{rescue}-T6. We show that iPSC-carrier-A1 symNs are not hyperactive, again puzzling, since it has the same genotype as iPSC-FD_{rescue}-T6, which is hyperactive. There are reports in the FD mouse that the dosage of *elp1* protein dictates the phenotypic severity¹³. Thus, we followed the hypothesis that *elp1* protein dosage underlies symN hyperactivity. **Sup. Fig 7e, f** as well as previous results¹¹ suggest that, for unknown reasons, higher *ELP1* protein levels in iPSC-carrier-A1 symNs compared to iPSC-FD_{rescue}-T6 symNs provide it with protection against hyperactivity.

Please see revised manuscript page 13, line 282 and **Sup. Fig 7**

To address this reviewers concern about the overall validity of our cellular model, we have now added new work that recapitulates previous findings from the mouse model of NGF retrograde transport defects in FD¹. **Sup. Fig. 8** shows that in our cellular model this NGF retrograde transport defect is conserved, supporting the overall validity of our model to represent disease aspects accurately.

Please see revised manuscript page 16, line 357 and **Sup. Fig 8**

- The authors reference the role of *ELP1* in translation, but *ELP1* also has a well-established role in transcriptional elongation. Is the hyperactive phenotype identified in these cell types potentially independent from or secondary to reduction of *ELP1*? Indeed, the *ELP1* mutation in FD has been linked to transcriptional elongation¹⁴⁻¹⁶, tRNA modification^{17,18}, and list more here especially vesicular transport^{19,20}, exocytosis²¹, and NGF retrograde signaling (endocytosis)¹. There are thoughts in the field that the many described elongator dysfunctions in the cell might all be downstream of the tRNA modification problem²². To address the intriguing question if the hyperactivity phenotype is downstream of the *ELP1* mutation and to establish a potential link between *ELP1* mutation and NET problems, we have now conducted further analyses. We assessed the literature and found that vesicle transport defects have been linked to FD^{1,19}. Furthermore, several studies indicate that RAB proteins are important in protein trafficking and might be implicated in FD^{19,23-28}. Importantly, Goffena et al., showed that proteins of a specific length and AA:AG codon ratio were most affected by elongator dysfunction. Re-analyzing their data from FD animal sensory neurons, we found that many RAB proteins fall into this category. Thus, we analyzed a panel of RAB proteins in our FD and control symNs and found that indeed RAB3A is differentially expressed in FD and it co-localizes with *ELP1* and NET (**Fig. 4m,n**). Lastly, we performed bulk RNA

seq. analysis comparing control and FD symNs and found a large number of RAB proteins dysregulated in FD (**Sup. Fig. 9**). Together, we believe that the *ELP1* mutation leads to defects in RAB proteins, leading to NET trafficking and problems of its expression on the cell surface, finally leading to symN hyperactivity.

Please see revised manuscript page 17, line 347, page 17, line 383, **Fig. 4m**, **Sup. Fig. 9**

- If this phenotype (hyperactivity) is the result of a mutation in a modifier gene, the authors should first demonstrate that the mutation of the modifier gene is associated with disease severity, and then establish the link between this mutation and the hyperactive phenotype. This study is based on two FD lines that both happen to have a mutation in the *LAMB4* gene. How many FD patients have a mutation in the *LAMB4* gene? What is the frequency of this mutation in the general population and in the FD population? Did the authors correct the *ELP1* mutation also in the iPSC-FD-S3 cell line to confirm no phenotypic rescue? This is a critical finding that certainly needs further investigation in order to demonstrate that this specific phenotype is relevant for FD pathogenesis and not coincidental. As explained above, we believe that the analysis of the frequency of the modifier mutation in the FD population is very important, but outside the scope of the current work. As detailed in Zeltner et al., 2016 the *LAMB4* mutation frequency in the generally population is extremely low¹¹, yet the frequency of this mutation in the greater FD population still needs to be established. As explained above, we believe that this analysis, while very important for the FD field, is outside the scope of this revision and manuscript. In addition, we believe that this analysis would not be a good fit here and would rather distract and confuse the current story, which focuses on symN hyperactivity in FD. However, as detailed above in point 1 of reviewer 3, here, using mild FD lines, we ruled out that *LAMB4* is the culprit of symN hyperactivity.

Please see revised manuscript page 13, line 282, **Sup. Fig. 7**

- In the discussion, the authors speculate that a potential explanation for the no observed rescue of the symN hyperactivity after correction of the FD major mutation could be attributed to “the downstream consequence of altered developmentally-timed genetic programs due to the loss of *Elp1* expression and hence they cannot be re-set once the nervous system is mature.” The authors should clarify this point. Isn’t the mutation corrected at the iPSC stage? If yes, why are they referring to mature nervous system here? If not, they should clarify at which stage of the neuronal differentiation protocol the *ELP1* mutation was corrected. We agree with this reviewer that this wording is unclear and confusing. With our new data about RAB proteins and the exclusion that *LAMB4* plays a role in hyperactivity, we now have reworded the discussion accordingly.

Please see revised manuscript page 24, line 541

Minor comments:

- In the Abstract (lines 21-22), the authors state: “Familial dysautonomia (FD) is a rare neurodevelopmental and neurodegenerative disorder that affects the sympathetic nervous system.” More precisely FD affects both the sensory and sympathetic nervous system. This is corrected now.

Please see revised manuscript page 1, line 23

- In the Introduction (line 42), the word “complicated” should be replaced with “complex” since this adjective is more appropriate in this context. This is adjusted now.

Please see revised manuscript page 2, line 44

- In the introduction (lines 77-78 and lines 152-153) there are citations missing that describe the characterization of sympathetic ganglia volume in mouse models of FD, several studies have been published. References were added.
Please see revised manuscript page 4, line 80
- The authors should use correct capitalization to distinguish human from mouse, ELP1, Elp1. Thank you for pointing this out, we now adjusted the nomenclature throughout the manuscript.
- In line 266 one bracket is missing. This was corrected.
Page 14, line 419

References

1. Li, L., Gruner, K. & Tourtellotte, W.G. Retrograde nerve growth factor signaling abnormalities in familial dysautonomia. *J Clin Invest* **130**, 2478-2487 (2020).
2. Stemper, B., *et al.* Sympathetic and parasympathetic baroreflex dysfunction in familial dysautonomia. *Neurology* **63**, 1427-1431 (2004).
3. Bar-Aluma, B.E. Familial Dysautonomia. in *GeneReviews((R))* (eds. Adam, M.P., *et al.*) (Seattle (WA), 1993).
4. Hilz, M.J., *et al.* Cold face test demonstrates parasympathetic cardiac dysfunction in familial dysautonomia. *Am J Physiol* **276**, R1833-1839 (1999).
5. Bremner, F.D. & Smith, S.E. Pupil abnormalities in selected autonomic neuropathies. *J Neuroophthalmol* **26**, 209-219 (2006).
6. Ernsberger, U. & Rohrer, H. Sympathetic tales: subdivisions of the autonomic nervous system and the impact of developmental studies. *Neural Dev* **13**, 20 (2018).
7. Jackson, M.Z., Gruner, K.A., Qin, C. & Tourtellotte, W.G. A neuron autonomous role for the familial dysautonomia gene ELP1 in sympathetic and sensory target tissue innervation. *Development* **141**, 2452-2461 (2014).
8. George, L., *et al.* Familial dysautonomia model reveals Ikbkap deletion causes apoptosis of Pax3+ progenitors and peripheral neurons. *Proc Natl Acad Sci U S A* **110**, 18698-18703 (2013).
9. Pearson, J. & Pytel, B.A. Quantitative studies of sympathetic ganglia and spinal cord intermedio-lateral gray columns in familial dysautonomia. *J Neurol Sci* **39**, 47-59 (1978).
10. Norcliffe-Kaufmann, L., Slangenaupt, S.A. & Kaufmann, H. Familial dysautonomia: History, genotype, phenotype and translational research. *Prog Neurobiol* **152**, 131-148 (2017).
11. Zeltner, N., *et al.* Capturing the biology of disease severity in a PSC-based model of familial dysautonomia. *Nat Med* **22**, 1421-1427 (2016).
12. Chan, J.W. & Teo, A.K.K. Replicates in stem cell models-How complicated! *Stem Cells* **38**, 1055-1059 (2020).
13. Dietrich, P., Alli, S., Shanmugasundaram, R. & Dragatsis, I. IKAP expression levels modulate disease severity in a mouse model of familial dysautonomia. *Hum Mol Genet* **21**, 5078-5090 (2012).
14. Otero, G., *et al.* Elongator, a multisubunit component of a novel RNA polymerase II holoenzyme for transcriptional elongation. *Mol Cell* **3**, 109-118 (1999).
15. Pokholok, D.K., *et al.* Genome-wide map of nucleosome acetylation and methylation in yeast. *Cell* **122**, 517-527 (2005).
16. Wittschieben, B.O., *et al.* A novel histone acetyltransferase is an integral subunit of elongating RNA polymerase II holoenzyme. *Mol Cell* **4**, 123-128 (1999).

17. Esberg, A., Huang, B., Johansson, M.J. & Bystrom, A.S. Elevated levels of two tRNA species bypass the requirement for elongator complex in transcription and exocytosis. *Mol Cell* **24**, 139-148 (2006).
18. Mehlgarten, C., *et al.* Elongator function in tRNA wobble uridine modification is conserved between yeast and plants. *Mol Microbiol* **76**, 1082-1094 (2010).
19. Lefler, S., *et al.* Familial Dysautonomia (FD) Human Embryonic Stem Cell Derived PNS Neurons Reveal that Synaptic Vesicular and Neuronal Transport Genes Are Directly or Indirectly Affected by IKBKAP Downregulation. *PLoS One* **10**, e0138807 (2015).
20. Cheishvili, D., *et al.* IKAP/Elp1 involvement in cytoskeleton regulation and implication for familial dysautonomia. *Hum Mol Genet* **20**, 1585-1594 (2011).
21. Rahl, P.B., Chen, C.Z. & Collins, R.N. Elp1p, the yeast homolog of the FD disease syndrome protein, negatively regulates exocytosis independently of transcriptional elongation. *Mol Cell* **17**, 841-853 (2005).
22. Close, P., *et al.* Transcription impairment and cell migration defects in elongator-depleted cells: implication for familial dysautonomia. *Mol Cell* **22**, 521-531 (2006).
23. Ahmed, B.A., *et al.* Serotonin transamidates Rab4 and facilitates its binding to the C terminus of serotonin transporter. *J Biol Chem* **283**, 9388-9398 (2008).
24. Eriksen, J., Bjorn-Yoshimoto, W.E., Jorgensen, T.N., Newman, A.H. & Gether, U. Postendocytic sorting of constitutively internalized dopamine transporter in cell lines and dopaminergic neurons. *J Biol Chem* **285**, 27289-27301 (2010).
25. Matthies, H.J., *et al.* Rab11 supports amphetamine-stimulated norepinephrine transporter trafficking. *J Neurosci* **30**, 7863-7877 (2010).
26. Mroczek, M., Kabzinska, D. & Kochanski, A. Molecular pathogenesis, experimental therapy and genetic counseling in hereditary sensory neuropathies. *Acta Neurobiol Exp (Wars)* **75**, 126-143 (2015).
27. Saxena, S., Bucci, C., Weis, J. & Kruttgen, A. The small GTPase Rab7 controls the endosomal trafficking and neuritogenic signaling of the nerve growth factor receptor TrkA. *J Neurosci* **25**, 10930-10940 (2005).
28. Tourtellotte, W.G. Axon Transport and Neuropathy: Relevant Perspectives on the Etiopathogenesis of Familial Dysautonomia. *Am J Pathol* **186**, 489-499 (2016).

REVIEWER COMMENTS

Reviewer #1 (Remarks to the Author):

All the raised concerns were properly addressed.

Reviewer #2 (Remarks to the Author):

The authors have addressed all my comments appropriately.

Reviewer #4 (Remarks to the Author) (note reviewer 4 co-reviewed with reviewer 3 on the previous round, and submitted their joint report here).

The authors have improved the manuscript considerably by including additional data and expanding on their descriptions and analysis. Overall, the response of the authors to the initial review comments appears satisfactory, but there are some critical points that need to be addressed in order to strengthen the validity of their cellular model.

Major comments:

1) In the Results (lines 285-287), the authors state: "Since this was a somewhat puzzling result, we employed more FD and control iPSC lines (Sup. Fig 7b) and assessed their electric activity in symNs (Sup. Table 3, Sup. Fig. 7c, d). These lines have various, but defined genotypes in ELP1 and LAMB4 (Sup. Table 3)." It is not clear what the authors mean by "various, but defined genotypes". It would be helpful to include more information on their genotypes in the main manuscript. Also, the name of the cell line in the figures and text does not match the name of the line in Supplementary Table 3: "iPSC-ctrl-652" vs "iPSC-ctrl-658.4b".

2) It is puzzling that the het CRISPR rescued line does not express the same amount of ELP1 protein as the carrier line. Both lines should carry the same ELP1 alleles: one ELP1 WT allele and one allele with the FD major splicing mutation. However, this represents the key finding explaining why there is no rescue of symN hyperactivity in the CRISPR line. In the discussion, the authors should provide a justification for this unexpected result (low level of ELP1 protein in the CRISPR rescued line vs. carrier line).

3) In the Results (lines 435-437), the authors state: "As expected, both drugs did not alter wild-type ELP1 splicing levels (Fig. 5j). This result is consistent with our earlier finding that genetic reversal of the ELP1 mutation does not rescue symN hyperactivity (Sup. Fig. 7a)." This statement is inconsistent with the newly added data explaining why the so-called "genetic reversal" did not rescue the symN hyperactivity. The author concluded that there is no rescue of symN hyperactivity in the CRISPR lines because the ELP1 amount is still low, not because ELP1 is not involved in this phenotype. The fact that drugs do not alter ELP1 splicing can simply be explained by an ELP1-independent mechanism of action. These drugs might act downstream or independently from ELP1 pathway. Please revise this statement.

Minor comments:

1) In the Introduction (lines 80), there are still some missing citations when referring to the size reduction of sympathetic ganglia in mice. Please include Morini et al. HMG 2016 and Dietrich et al. HMG 2012.

2) The “E” of the Elongator complex has to be capitalized.

3) In the Results (lines 293-294), the authors state: “iPSC-FD-mild-M4 is a genotypically mild FD line that we reprogrammed here.” The authors should replace “genotypically” with “phenotypically”. The ELP1 genotype is the same for all the FD lines.

Point-by-point rebuttal for Wu et al., Nature Communications:

We are very excited and thankful that the reviewers liked our revision and manuscript. We have now addressed all outstanding requests in the manuscript and figures and such changes were highlighted in red font in the re-revised manuscript and the below point-by-point rebuttal for easy reference. We also list the page and line numbers, where specific changes can be found in the re-revised manuscript.

Reviewer #1:

All the raised concerns were properly addressed. Thank you.

Reviewer #2:

The authors have addressed all my comments appropriately. Thank you.

Reviewer #3 and 4:

The authors have improved the manuscript considerably by including additional data and expanding on their descriptions and analysis. Overall, the response of the authors to the initial review comments appears satisfactory, but there are some critical points that need to be addressed in order to strengthen the validity of their cellular model.

We appreciate these additional comments and have them addressed one-by-one below.

Major comments:

1) In the Results (lines 285-287), the authors state: "Since this was a somewhat puzzling result, we employed more FD and control iPSC lines (Sup. Fig 7b) and assessed their electric activity in symNs (Sup. Table 3, Sup. Fig. 7c, d). These lines have various, but defined genotypes in ELP1 and LAMB4 (Sup. Table 3)." It is not clear what the authors mean by "various, but defined genotypes". It would be helpful to include more information on their genotypes in the main manuscript. We have expanded on that, page 13, line 282. Also, the name of the cell line in the figures and text does not match the name of the line in Supplementary Table 3: "iPSC-ctrl-652" vs "iPSC-ctrl-658.4b". Thank you for that catch, we have fixed that typo now.

2) It is puzzling that the het CRISPR rescued line does not express the same amount of ELP1 protein as the carrier line. Both lines should carry the same ELP1 alleles: one ELP1 WT allele and one allele with the FD major splicing mutation. However, this represents the key finding explaining why there is no rescue of symN hyperactivity in the CRISPR line. In the discussion, the authors should provide a justification for this unexpected result (low level of ELP1 protein in the CRISPR rescued line vs. carrier line differs). We have addressed this now in the discussion, page 23, line 516.

3) In the Results (lines 435-437), the authors state: "As expected, both drugs did not alter wild-type ELP1 splicing levels (Fig. 5j). This result is consistent with our earlier finding that genetic reversal of the ELP1 mutation does not rescue symN hyperactivity (Sup. Fig. 7a)." This statement is inconsistent with the newly added data explaining why the so-called "genetic reversal" did not rescue the symN hyperactivity. The author concluded that there is no rescue of symN hyperactivity in the CRISPR lines because the ELP1 amount is still low, not because ELP1 is not involved in this phenotype. The fact that drugs do not alter ELP1 splicing can simply

be explained by an ELP1-independent mechanism of action. These drugs might act downstream or independently from ELP1 pathway. Please revise this statement. **We have modified this statement now. Page 19, line 432.**

Minor comments:

1) In the Introduction (lines 80), there are still some missing citations when referring to the size reduction of sympathetic ganglia in mice. Please include Morini et al. HMG 2016 and Dietrich et al. HMG 2012. **We included these citations now. Page 4, line 75.**

2) The “E” of the Elongator complex has to be capitalized. **Fixed, page 2, line 44, 45.**

3) In the Results (lines 293-294), the authors state: “iPSC-FD-mild-M4 is a genotypically mild FD line that we reprogrammed here.” The authors should replace “genotypically” with “phenotypically”. The ELP1 genotype is the same for all the FD lines. **Fixed, page 13, line 291.**